# PUSHING THE LIMITS OF PRE-TRAINING FOR TIME SERIES FORECASTING IN THE CLOUDOPS DOMAIN

## ABSTRACT

Time series has been left behind in the era of pre-training and transfer learning. While research in the fields of natural language processing and computer vision are enjoying progressively larger datasets to train massive models, the most popular time series datasets consist of only tens of thousands of time steps, limiting our ability to study the effectiveness of pre-training and scaling. Recent studies have also cast doubt on the need for expressive models and scale. To alleviate these issues, we introduce three large-scale time series forecasting datasets from the cloud operations (CloudOps) domain, the largest having billions of observations, enabling further study into pre-training and scaling of time series models. We build the empirical groundwork for studying pre-training and scaling of time series models and pave the way for future research by identifying a promising candidate architecture. We show that it is a strong zero-shot baseline and benefits from further scaling, both in model and dataset size. Accompanying these datasets and results is a suite of comprehensive benchmark results comparing classical and deep learning baselines to our pre-trained method – achieving a 27% reduction in error on the largest dataset.

## 1 INTRODUCTION

Pre-training and transfer learning has enabled the next generation of advances in machine learning. From large language models (LLMs) trained on web-scale data (Brown et al., 2020) subsequently yielding chatbots (Touvron et al., 2023) and autonomous agents (Park et al., 2023), to generative models capable of creating images and videos based on text descriptions (Rombach et al., 2022). Pre-training, the predominant approach to transfer learning, has allowed us to learn general representations on large-scale datasets, subsequently adapting to downstream datasets and tasks. Striking capabilities in performance and generalization, such as zero-shot capabilities and in-context learning, arise with the key ingredient of scaling model and pre-training data size.

While transfer learning for time series forecasting has been explored in the form of multi-task learning (Semenoglou et al., 2021; Benidis et al., 2022; Nie et al., 2023), pre-training has not yet received significant attention. Firstly, there is currently a lack of large-scale public domain time series data available to fuel the pre-training of large time series models – the most widely adopted time series forecasting datasets consists of only tens of thousands of time steps (Wu et al., 2021; Salinas et al., 2020). While vasts quantities of time series data are generated everyday, access to such data is typically restricted to their respective owners. We argue that small-scale academic datasets bring conflicting evidence regarding the need for scale and expressive models in time series forecasting (Makridakis et al., 2018; Zeng et al., 2023; Xu et al., 2023). For instance, Zeng et al. (2023) highlighted that lightweight models outperform expressive Transformer-based architectures and data scale is not a limiting factor. Secondly, unlike image and text data which naturally share semantic information across datasets and domains, time series data may not enjoy such properties of transferability as the semantics of time series data may be unique to their dataset or domain. As such, it is still unclear how time series models can benefit from pre-training and transfer learning.

To address this issue, we first provide definitions of time-series transferability at various degrees of granularity. As illustrated in Figure 1, time series across different domains only share the knowledge of generic time series concepts. Collections from the same domain share some domain knowledge but typically face a larger degree of distribution shift, and even face the problem of heterogeneity–different collections have varying dimensionality, sampling frequencies, and covariates. Finally,

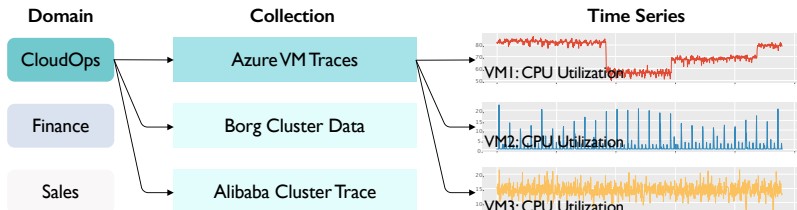

Figure 1: Hierarchy of time series datasets. Time series can be classified into **domains** at the top level, which are broad application areas which generate time series with shared characteristics. Each domain consists of many **collections** of time series, which are defined to be sets of time series which measure semantically identical observations across different objects, i.e. in the Azure VM Traces collection, each time series measures the CPU utilization for different virtual machines, and also record the same covariates.

transferability between time series from the same collection is the highest, since they are typically generated by the same underlying system.

Next, we introduce three large-scale time series datasets from the cloud operations (CloudOps) domain. Cloud providers generate trillions of time series data points everyday. Forecasting these time series are critical for their daily operation tasks, ranging from anomaly detection (Hochenbaum et al., 2017) to resource allocation (Luo et al., 2020), and many other tasks (Cheng et al., 2023). CloudOps is well positioned to benefit from pre-trained time series models, whether it be simply from improved performance and generalization, or even for cold-start problems (Fatemi et al., 2023).

Based on these three datasets, we focus on pre-training within the same collection, known as the in-collection setting, systematically studying the various components of the forecasting pipeline and their scaling capabilities. Preliminary results indicate that pre-trained models in the in-collection setting are strong zero-shot forecasters, prompting us to focus on zero-shot transfer. We then propose a recipe for building a powerful and scalable pre-trained time series model, establishing a strong zero-shot baseline. Specifically, (1) we find that the masked encoder Transformer architecture provides superior performance and cost trade-offs compared to existing Transformer variants, (2) a Student-T parametric distribution output head performs robustly across datasets, and (3) date/time features are insufficient in providing positional information – positional encodings are critical. Finally, we study the scaling behavior of these methods and show promising results which indicate towards further scaling of model and data size. We summarize our contributions in the following:

- We introduce three large-scale CloudOps time series forecasting datasets, enabling further study into pre-training and transfer learning for time series models, and perform a comprehensive benchmarking of classical and deep learning baselines.

- We perform a series of experiments, building the empirical groundwork to study pre-training and scaling, paving the way for future work in the field of pre-training for time series forecasting. Our candidate architecture, when pre-trained, beats the aforementioned baselines as a zero-shot forecaster in the in-collection setting.

- We show that time series models benefit from scaling – on our largest dataset, error reduces by 6% as we scale parameter count by 8x, and 9% when we scale dataset size by 100x.

## 2 SETUP

**Problem Formulation** Consider a dataset of $n$ time series $\mathcal{D} = \{(\boldsymbol{Y}^{(i)}, \boldsymbol{Z}^{(i)})\}_{i=1}^{n}$, where $\boldsymbol{Y}^{(i)} = (\boldsymbol{y}_1^{(i)}, \ldots, \boldsymbol{y}_{T_i}^{(i)})$ is a time series of $T_i$ time steps, and $\boldsymbol{y}_t^{(i)} \in \mathbb{R}^{d_y}$ is the target value at the $t$-th time step of the $i$-th time series. Each time series has an associated set of covariates, $\boldsymbol{Z}^{(i)} = (\boldsymbol{z}_1^{(i)}, \ldots, \boldsymbol{z}_{T_i}^{(i)})$, where $\boldsymbol{z}_t^{(i)} \in \mathbb{R}^{d_z}$. The range $(1, \ldots, \tau_i)$ is considered to be the training set, and $(\tau_i + 1, \ldots, T_i)$ to be the test set. We are interested in the time series forecasting task of predicting the conditional distribution,

$$p(\boldsymbol{Y}_{t+1:t+H}^{(i)} | \boldsymbol{Y}_{1:t}^{(i)}, \boldsymbol{Z}_{1:t+H}^{(i)}; \boldsymbol{\theta}), \ \forall t = \tau_i, \tau_i + s, \ldots, \tau_i + ms,$$

where $H$ is the prediction length or forecast horizon, $s$ is the stride at test time, and $m$ is the largest non-negative integer such that $\lceil (\tau_i + ms)/(T_i - H) \rceil = 1$. This conditional distribution is

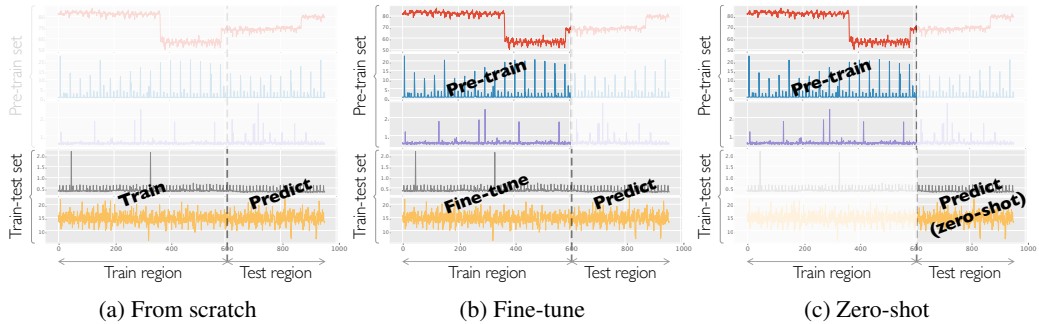

Figure 2: **In-collection pre-training** adaptation strategies. Given a collection of time series, the collection is split into two non-overlapping subsets of time series, known as the pre-train and train-test sets. A model pre-trained on the pre-train set can then be adapted for making predictions on the held-out test region of the train-test set, either by undergoing further fine-tuning, or via zero-shot predictions.

parameterized as a neural network, with $\boldsymbol{\theta}$ as its parameters. We focus on the in-collection setting, illustrated in Figure 2. Here, we further consider a pre-training dataset, $\mathcal{D}_{\mathrm{pt}} = \{(\boldsymbol{Y}_{1:\tau_i}^{(i)}, \boldsymbol{Z}_{1:\tau_i}^{(i)})\}_{i=n+1}^{n+n_{\mathrm{pt}}}$, on which the model is pre-trained, but not evaluated on.

## 2.1 DATASETS

To support the evaluation of transferability in pre-trained time series models, a sufficiently data-rich pre-training dataset is required. We bridge this gap by introducing three datasets from the CloudOps domain, ranging from 100 million to a billion observations (where $\#\mathrm{obs.} = \sum_{i=1}^{n+n_{\mathrm{pt}}} T_i$), with key statistics summarized in Table 1. We pre-processed these datasets from traces of large compute clusters which have been made publicly available, into time series format. Below are brief descriptions of the datasets, with full details in Appendix A.

Table 1: Key statistics of CloudOps time series forecasting datasets.

| Dataset | # Time Series | | Length | | | | Total Obs. | # Tgts. | Freq. |
| | Pre-train | Train-test | Min. | Max. | Med. | Mean | | | |
| --- | --- | --- | --- | --- | --- | --- | --- | --- | --- |
| azure2017 | 159,472 | 17,568 | 673 | 8,640 | 8,640 | 6,117 | 1,082,965,986 | 1 | 5min. |
| borg2011 | 143,386 | 11,117 | 674 | 8,352 | 2,911 | 4,321 | 667,657,472 | 2 | 5min. |
| ali2018 | 58,409 | 6,048 | 676 | 2,304 | 2,304 | 2,199 | 141,762,617 | 2 | 5min. |

- The **Azure VM Traces 2017** (azure2017) dataset (Cortez et al., 2017) is a representative subset of first-party (internal) Azure virtual machine (VM) workloads collected in 2017 in one geographical region (anonymized). The main performance metric monitored in the raw dataset is the CPU utilization. We have pre-processed this into a univariate forecasting dataset, in which we are interested to predict the average CPU utilization over 48 time steps in 5-minute intervals.

- The **Borg Cluster Data 2011** (borg2011) dataset (Wilkes, 2011) represents 29 days worth of Borg (Google cluster manager) cell information in May 2011 on a cluster of 12.5k machines. This is a multivariate forecasting dataset, in which we are interested in predicting the CPU rate and canonical memory usage over 48 time steps in 5-minute intervals.

- The **Alibaba Cluster Trace 2018** (ali2018) dataset (Guo et al., 2019) is a collection from a cluster of around 4000 machines over 8 days. This is a multivariate forecasting dataset, in which we are interested in predicting the CPU utilization percentage and memory utilization percentage over 48 time steps in 5-minute intervals.

Following the collection of these large-scale datasets, we define a split to divide them into a pre-train set for pre-training, and a train-test set for the downstream task. For each dataset, we perform a roughly 90/10 split (by time series) into pre-train and train-test sets, respectively. As some time series can be related (e.g. a VM can belong to the same user or be running the same task), we perform the split based on the top level attribute to ensure that there is no data leakage between the pre-train

and train-test sets. Furthermore, all time series in the train-test set are end-aligned (all ending on the same date/time). Finally, we also highlight that pre-training is not performed on the time period corresponding to the test set. A more detailed description of this pre-processing step can be found in Appendix A.2.

## 2.2 PRE-TRAINING & DOWNSTREAM TASK

**Pre-training Task**    While self-supervised objectives have been introduced for time series forecasting (Yue et al., 2022; Woo et al., 2022a), our focus lies in architecture design and scaling capabilities, thus we focus on supervised pre-training. The loss function may vary depending on the probabilistic forecasting head used, e.g. negative log-likelihood for parametric distributions, or (weighted) quantile losses for quantile functions.

**Downstream Task**    We focus on the in-collection setting for the forecasting task, highlighting two adaptation strategies amenable with the supervised pre-training task. Illustrated in Figure 2, we can further fine-tune the pre-trained model, or directly leverage it for zero-shot predictions. For purposes of evaluation, we construct a train-test split, defining the test set to be the last 12 non-overlapping windows of horizon length 48, i.e. $H = s = 48, m = 12$, and the train set being everything before that. Both point and probabilistic forecasts are evaluated. Point forecasts are evaluated with the symmetric mean absolute percentage error (sMAPE). For the multivariate setting, it is simply averaged across dimensions. Probabilistic forecasts are evaluated with the Continuous Ranked Probability Score (CRPS) and CRPS-sum metrics for univariate and multivariate datasets respectively. Further details on evaluation metrics can be found in Appendix B.

## 2.3 MODELS

In order to benefit from pre-training, we need to identify an architecture with strong scaling capabilities. Transformers (Vaswani et al., 2017) have emerged as a powerful general architecture, capable of modelling a variety of data modalities, as well as having the capability of scaling massively, to trillions of data observations and billions of parameters. While there has been a variety of time series specific Transformer architectures, modifying the attention mechanism or layer structure to incorporate time series specific inductive biases such as seasonal-trend decomposition (Wu et al., 2021; Zhou et al., 2022; Woo et al., 2022b) , our focus instead lies in studying a simple and scalable method's capabilities in the pre-training setting. Thus, we focus on the original scaled dot product attention, and consider several Transformer variants which have been shown to be effective in the time series setting (Section 4.2).

## 3  RELATED WORK

**Pre-train + Fine-tune**    Pre-training for time series forecasting (Ma et al., 2023) has previously been limited due to dataset limitations. Yue et al. (2022); Woo et al. (2022a) consider a two-stage approach to forecasting, first performing self-supervised pre-training stage to learn representations, then a supervised learning stage for the downstream (forecasting) task. However, due to dataset limitations, they only study the setting where the pre-training and downstream task was performed on the same set of time series, and thus did not study the transfer learning capabilities of such methods. The idea of leveraging LLMs pre-trained on text data as an initialization to subsequently fine-tune on time series data has been explored recently (Zhou et al., 2023; Chang et al., 2023). Such methods try to address the lack of large-scale time series data for pre-training by leveraging data from other modalities.

**Zero-shot Forecasting**    Zero-shot transfer has been explored for time series forecasting, with focus on the univariate setting. Oreshkin et al. (2021) initially showed that the N-BEATS model (Oreshkin et al., 2020) implicitly performed meta-learning updates, allowing an ensemble of models to achieve good performance when the source and target datasets came from the same domain (M4 and FRED, both economics), but still subpar to training directly on the target dataset. Khurana et al. (2023) introduces a zero-shot forecasting model trained purely on synthetic data. They introduce a synthetic data distribution inspired by real world time series, and show that the proposed method performs well on low resource settings.

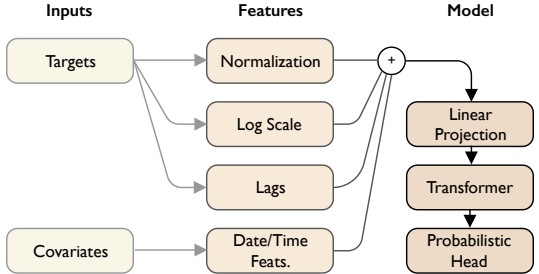

Figure 3: Data flow of a standard pipeline for probabilistic time series forecasting (Salinas et al., 2020). Time series datasets typically consists of the target time series and covariates (additional features such as date/time information, and auxiliary time series). The target time series is normalized, and features are extracted – log scale and lags from the targets, and date/time features from the covariates. The targets, extracted features, and covariates are concatenated, before being fed into the model.

Table 2: Results of baseline methods on the validation set, averaged over 5 independent (pre-)training runs, with standard deviations in brackets. Results for a model trained from scratch (without pre-training) on the train set is reported for comparison.

|  | azure2017 | | borg2011 | | ali2018 | |
| --- | --- | --- | --- | --- | --- | --- |
|  | sMAPE | CRPS | sMAPE | $\text{CRPS}_{\text{sum}}$ | sMAPE | $\text{CRPS}_{\text{sum}}$ |
| Fine-tuning | $0.100_{\pm 0.001}$ | $0.095_{\pm 0.001}$ | $0.076_{\pm 0.013}$ | $0.138_{\pm 0.013}$ | $0.166_{\pm 0.001}$ | $0.020_{\pm 0.000}$ |
| Zero-shot | $0.100_{\pm 0.001}$ | $0.095_{\pm 0.000}$ | $0.070_{\pm 0.007}$ | $0.130_{\pm 0.009}$ | $0.166_{\pm 0.001}$ | $0.020_{\pm 0.000}$ |
| No pre-training | $0.140_{\pm 0.011}$ | $0.129_{\pm 0.007}$ | $0.085_{\pm 0.014}$ | $0.159_{\pm 0.012}$ | $0.267_{\pm 0.031}$ | $0.080_{\pm 0.018}$ |

## 4 EXPERIMENTS

Starting from a standard approach to forecasting with Transformers in Section 4.1 to form a reasonable baseline, we show that for the in-collection setting, pre-trained models are strong zero-shot forecasters. We then examine various Transformer based schemes for forecasting in Section 4.2, and perform a series of ablations to construct a strong zero-shot baseline culminating in a comprehensive benchmark against classical and deep learning methods in Section 4.3. We further study this scaling behavior in Section 4.4. Our results build the empirical groundwork for scaling these general architectures, shedding light on the flexibility and various tradeoffs these models make.

### 4.1 BASELINE

A standard pipeline for time series forecasting is described in Figure 3. Features are extracted from the input data, which is then fed into a pipeline comprising a linear projection (mapping from observation/feature space into representation space), the Transformer model, and a probabilistic head.

**Model** Our baseline Transformer is the canonical encoder-decoder architecture (Vaswani et al., 2017), performing **iterative multi-step (IMS)** decoding. Illustrated in Figure 4, the encoder takes the targets and covariates of the context window as inputs, and the decoder takes as input the lagged target and covariates, $(\boldsymbol{y}_{t-1}, \boldsymbol{z}_t)$, of the prediction horizon as inputs to predict $\boldsymbol{y}_t$. This base model size has 6 encoder and decoder layers with a hidden size of $d_{\text{model}} = 384$. Further details in Appendix C.1.

**Training** We pre-train the models over 100,000 iterations with a batch size of 512, yielding a total of 51,200,000 samples. Each sample is obtained by first randomly selecting a time series in proportion to each time series length, then uniformly sampling a window of length $L + H$, where $L = 480$ is the lookback window or context length. We use the AdamW (Loshchilov & Hutter, 2019) optimizer with a learning rate of 1e-3, performing linear warm up over 10,000 steps, and cosine annealing subsequently. Fine-tuning/training from scratch methodology is similar to the methodology in benchmark (Appendix E.2) except that we search over 5 learning rates, {1e-4, 2e-4, 5e-4, 1e-3, 2e-3} for no pre-training, and {1e-3, 1e-4, 1e-5, 1e-6, 1e-7} for fine-tuning, with a validation set, defined as the last horizon in the training set.

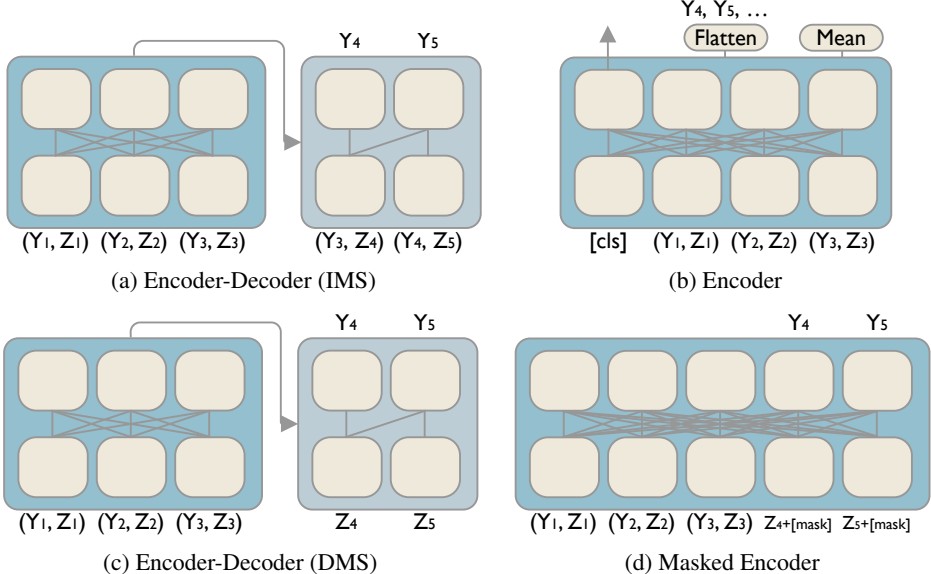

Figure 4: Designs of Transformer architecture variants.

**Results**  Table 2 reports the results of our baseline method on three different settings. We find that pre-training outperforms a model trained from scratch, and surprisingly, further fine-tuning a pre-trained model yields no benefits over zero-shot forecasts. One potential reason for this is that the pre-training data is sufficiently diverse for generalization to the train-test set, and that fine-tuning sometimes requires careful design and hyperparameter tuning. Based on these results, we focus on exploring the zero-shot capabilities of pre-trained time series models. Due to computational constraints, we only obtain standard deviations on pre-trained models for this section, over 5 independent runs, and assume similar standard deviations for pre-trained models in subsequent sections.

## 4.2 ARCHITECTURES

As highlighted in Section 4.1, our baseline model follows the original encoder-decoder Transformer architecture with IMS decoding. Many Transformer variants have since been introduced, each having their individual pros and cons. In the following, we review the various architecture designs (see Figure 4), highlight their differences, and compare them in terms of performance and computational cost.

**Direct multi-step (DMS) encoder-decoders** were introduced in Zhou et al. (2021) to overcome the drawback of IMS decoding requiring $H$ forward passes through the decoder – even with caching of intermediate decoder outputs, this posed a significant computational burden when forecasting over long horizons. Instead of taking $(\boldsymbol{y}_{t-1}, \boldsymbol{z}_t)$ pairs as input, the DMS decoder takes as input $\boldsymbol{z}_t$ to predict $\boldsymbol{y}_t$.

**Encoder** architectures have also recently been shown to perform well for time series forecasting. Nie et al. (2023) introduced an encoder architecture which obtains the output representation by a Flatten operation – concatenating the representations of all time steps into a representation of dimension $d_{\mathrm{model}} * L$. We also consider two simple methods to obtain output representations – "mean", performing average-pooling on the context representations, and "cls", giving the model a learnable embedding as input, and taking the corresponding representation as the output, analogous to BERT's [cls] token.

**Masked encoders** have been shown to be effective for time series tasks (Drouin et al., 2022; Tashiro et al., 2021), performing masked reconstruction (Devlin et al., 2019), where the input is replaced with a learnable mask embedding combined with position information. Specifically, we perform masking only on the prediction range. This is a DMS approach since we can predict multiple masked time steps in a single forward pass.

**Computational Cost**  Apart from performance, we are also interested in the computational costs associated with each architecture variant, summarized in Table 3. We consider an equivalency between these variants in terms of the number of layers, denoted $N$. Of course, this leads to encoder-decoder architectures having an advantage in terms of parameter count, since they have separate encoder and decoder layers, leading to approximately $2P$ parameters when masked encoders and encoder models have $P$ parameters. Yet, both encoder-decoders and masked encoders have similar computational

Table 3: Results of various Transformer architecture variants on the validation set. Best results **bolded** and second best underlined. $D = d_{\mathrm{model}}$, shortened for brevity, and $C$ represents the output size per time step, typically the dimensionality of time series multiplied by number of parameters for the output distribution.

| | | | FLOPs | | | azure2017 | | borg2011 | | ali2018 | |
|---|---|---|---|---|---|---|---|---|---|---|---|
| | Layers | Params | Attn. | Output | Dec. Iters. | sMAPE | CRPS | sMAPE | CRPS$_{\mathrm{sum}}$ | sMAPE | CRPS$_{\mathrm{sum}}$ |
| Enc-Dec (IMS) | $N$ | $2P$ | $N(L+H)^2$ | $CDH$ | $H$ | 0.100 | 0.095 | 0.070 | 0.130 | 0.166 | 0.020 |
| Enc-Dec (DMS) | $N$ | $2P$ | $N(L+H)^2$ | $CDH$ | 1 | 0.117 | 0.109 | 0.056 | 0.116 | **0.149** | **0.016** |
| Encoder (mean) | $N$ | $P$ | $NL^2$ | $CDH$ | 1 | 0.108 | 0.104 | 0.054 | 0.117 | 0.153 | **0.016** |
| Encoder (cls) | $N$ | $P$ | $NL^2$ | $CDH$ | 1 | 0.110 | 0.107 | 0.054 | 0.116 | 0.152 | **0.016** |
| Encoder (flatten) | $N$ | $P$ | $NL^2$ | $CDLH$ | 1 | 0.097 | 0.094 | 0.053 | 0.120 | **0.149** | **0.016** |
| Masked Encoder | $N$ | $P$ | $N(L+H)^2$ | $CDH$ | 1 | **0.094** | **0.093** | **0.052** | **0.115** | **0.149** | **0.016** |

costs in terms of number of floating point operations (FLOPs), in the order of $N(L + H)^2$, since encoder-decoders perform self-attention on both the context and prediction range individually, as well as cross-attention, whereas masked encoders perform self-attention over the context and prediction range combined. encoder models pose another trade-off – while they are more efficient in terms of a lower complexity in the attention layers, the flatten operation leads to a large projection layer.

**Results**    Table 3 reports performance on all Transformer variants. Our first observation is that although encoder-decoder models have higher parameter counts, they did not outperform masked encoder or the encoder models. Amongst encoder models, the flatten method outperforms the "mean" and "cls" approaches, likely due to the much larger representation and parameter size at the output head. Finally, we observe a close contest between masked encoders and flatten encoder models, neither significantly outperforming the other in any dataset. A direct comparison based on computation cost is also challenging, since both have their own pros and cons – masked encoders have a larger cost in their attention layers while flatten encoder models have larger cost in the output head. Ultimately, we select masked encoders due to firstly, having a lower parameter count – the output head of flatten encoder models have $CDLH$ parameters compared to $CD$ for masked encoders, and secondly, encoder models are less flexible, having a fixed input length, whereas we would want to consider variable input length in future work.

## 4.3 CLOUDOPS TIME SERIES FORECASTING BENCHMARK

### 4.3.1 A STRONG ZERO-SHOT BASELINE

By performing further ablation studies, we establish a recipe for a generic Transformer architecture to be pre-trained as a strong zero-shot forecaster. Due to space constraints, we summarize our key findings in the following, with full details available in Appendix D.

**Probabilistic Head**    Probabilistic forecasting requires a predictive probability distribution rather than just a point forecast. We compared the parametric distribution approach (Student-T) with several quantile function and normalizing flow based heads, and found that taking the simple approach of a parametric distribution proved to be a simple and robust choice, performing well across datasets and metrics, without any additional hyperparameter tuning.

**Positional Encoding**    Transformers rely on the attention mechanism which is permutation equivariant, requiring positional encodings to encode positional information. Time series has a natural approach, which is to leverage date/time features (e.g. minute-of-hour, day-of-week). We studied the impact of leveraging these features to encoding positional information, versus widely used approaches in the Transformer literature. We find that date/time information are not critical features for forecasting, and recommend the usage of a positional encoding method, using RoPE specifically.

**Scaling Up**    Finally, we scaled up our zero-shot recipe, training on three model sizes – Base, Large, and xLarge. Table 4 lists the specific hyperparameters used for each size. Along with scaling up the model size, we pre-train these models longer, for 400,000 iterations.

Table 4: Details of zero-shot model sizes.

| | Layers | $d_{\mathrm{model}}$ | $d_{\mathrm{ff}}$ | # heads | $d_{\mathrm{kv}}$ | # params |
|---|---|---|---|---|---|---|
| Base | 6 | 384 | 1536 | 6 | 64 | 10.7m |
| Large | 9 | 512 | 2048 | 8 | 64 | 28.4m |
| xLarge | 12 | 768 | 3072 | 12 | 64 | 85.1m |

Table 5: CloudOps benchmark. Results on various statistical, deep learning, and pre-trained baselines on the test set. Best results are **bolded** and second best underlined. "-" indicates that the method is only available for the univariate/multivariate setting. AutoETS returns exploding prediction intervals for many time series, thus we omit its results.

| | azure2017 | | borg2011 | | ali2018 | |
|---|---|---|---|---|---|---|
| | sMAPE | CRPS | sMAPE | CRPS$_{sum}$ | sMAPE | CRPS$_{sum}$ |
| Naive | 0.191 | 0.506 | 0.082 | 0.292 | **0.153** | 0.033 |
| AutoARIMA | 0.446 | 0.247 | - | - | - | - |
| AutoETS | 0.217 | - | - | - | - | - |
| AutoTheta | 0.361 | 0.290 | - | - | - | - |
| VAR | - | - | 0.096 | 0.135 | 0.189 | 0.024 |
| MQ-CNN | $0.258_{\pm0.035}$ | $0.186_{\pm0.002}$ | - | - | - | - |
| NBEATS | 0.149 | 0.134 | - | - | - | - |
| TFT | $0.142_{\pm0.002}$ | $0.121_{\pm0.000}$ | - | - | - | - |
| DeepAR | $0.110_{\pm0.001}$ | $0.101_{\pm0.001}$ | $0.103_{\pm0.006}$ | $0.135_{\pm0.001}$ | $0.168_{\pm0.002}$ | $0.020_{\pm0.000}$ |
| Autoformer | $0.221_{\pm0.023}$ | $0.216_{\pm0.017}$ | $0.112_{\pm0.009}$ | $0.191_{\pm0.001}$ | $0.206_{\pm0.006}$ | $0.032_{\pm0.002}$ |
| FEDformer | $0.175_{\pm0.010}$ | $0.189_{\pm0.004}$ | $0.114_{\pm0.012}$ | $0.191_{\pm0.002}$ | $0.200_{\pm0.003}$ | $0.034_{\pm0.001}$ |
| NSTransformer | $0.134_{\pm0.006}$ | $0.117_{\pm0.005}$ | $0.069_{\pm0.001}$ | $0.137_{\pm0.002}$ | $0.167_{\pm0.003}$ | $0.019_{\pm0.000}$ |
| PatchTST | $0.122_{\pm0.003}$ | $0.111_{\pm0.001}$ | $0.069_{\pm0.000}$ | $0.130_{\pm0.000}$ | $0.197_{\pm0.000}$ | $0.025_{\pm0.000}$ |
| LinearFamily | $0.202_{\pm0.002}$ | $0.163_{\pm0.004}$ | $0.091_{\pm0.003}$ | $0.188_{\pm0.001}$ | $0.174_{\pm0.001}$ | $0.024_{\pm0.000}$ |
| DeepTime | $0.208_{\pm0.000}$ | $0.184_{\pm0.001}$ | $0.085_{\pm0.004}$ | $0.183_{\pm0.006}$ | $0.194_{\pm0.001}$ | $0.030_{\pm0.001}$ |
| TimeGrad | - | - | $0.081_{\pm0.002}$ | $0.169_{\pm0.048}$ | $0.193_{\pm0.003}$ | $0.040_{\pm0.001}$ |
| TS2Vec | 0.190 | 0.156 | 0.113 | 0.159 | 0.231 | 0.045 |
| CoST | 0.189 | 0.151 | 0.118 | 0.161 | 0.229 | 0.045 |
| OFA | 0.140 | 0.120 | 0.069 | **0.124** | 0.164 | 0.020 |
| Meta N-BEATS | 0.120 | 0.116 | - | - | - | - |
| DeepAR-Base | 0.216 | 0.163 | 0.066 | 0.149 | 0.240 | 0.053 |
| Autoformer-Base | 0.171 | 0.165 | 0.187 | 0.235 | 0.266 | 0.065 |
| Ours-Base | 0.084 | 0.079 | 0.061 | 0.128 | 0.154 | **0.016** |
| Ours-Large | 0.081 | 0.078 | 0.061 | 0.129 | 0.155 | 0.017 |
| Ours-xLarge | **0.080** | **0.077** | **0.060** | 0.128 | 0.155 | 0.017 |

### 4.3.2 BENCHMARK

For classical methods, we compare against the naive method (Hyndman & Athanasopoulos, 2018), AutoARIMA, AutoETS, and AutoTheta (Hyndman & Khandakar, 2008; Garza et al., 2022) for univariate setting, and VAR (Seabold & Perktold, 2010) for multivariate. For deep learning models, we compare against probabilistic methods, MQ-CNN , Temporal Fusion Transformer (TFT), DeepAR, and TimeGrad (multivariate) and methods from the long sequence time series forecasting literature including Autoformer, FEDformer, NSTransformer, PatchTST, LinearFamily, and DeepTime. These methods follow the "from scratch" setting as per Figure 2. Finally, we compare with pre-training methods – DeepAR and Autoformer pre-trained and scaled to a similar size as "Base" (see Appendix E.3), as well as existing methods, TS2Vec (Yue et al., 2022), CoST (Woo et al., 2022a), Meta N-BEATS (Oreshkin et al., 2021), and One Fits All (OFA) (Zhou et al., 2023). Further training and hyperparameter details of baselines can be found in Appendix E.

**Results** Table 5 summarizes the results, reporting average and standard deviation for deep learning baselines over 3 independent runs. Statistical models are deterministic, and pre-trained methods (and N-BEATS, an ensemble of 18 models) are run once due to computational constraints. In the CloudOps domain with relatively high frequency data, the naive forecast acts is a strong baseline. Statistical methods which only learn from a single time series generally underperform even the naive forecast. However, deep learning based models which are global methods are generally stronger, of note are DeepAR and more recent methods such as NSTransformer and PatchTST. We hypothesize that Autoformer and FEDformer are underperforming even DeepAR due to time series from the CloudOps domain having high frequency with less focus on seasonality and trend features, favoring a general model architecture with fewer inductive biases.

Amongst the pre-trained methods, OFA surprisingly shows significant promise being adapted from a language model trained on text data. This idea could be pushed further, for example, by having a second stage pre-training on the time series pre-training data. We observe that our zero-shot approach constitutes a very strong baseline, obtaining a 27/24% reduction in sMAPE/CRPS from the next best performing method on the largest dataset, azure2017, generally outperforming all other methods. On a final note, we posit that the naive forecast is not the optimal prediction on the ali2018 dataset based

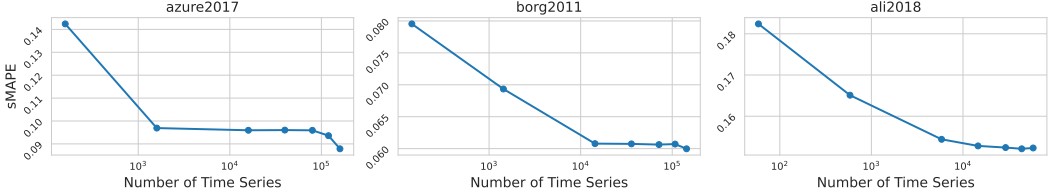

Figure 5: Performance curve against number of observations for pre-training across various model sizes. Each point represents the test performance on separate pre-training runs.

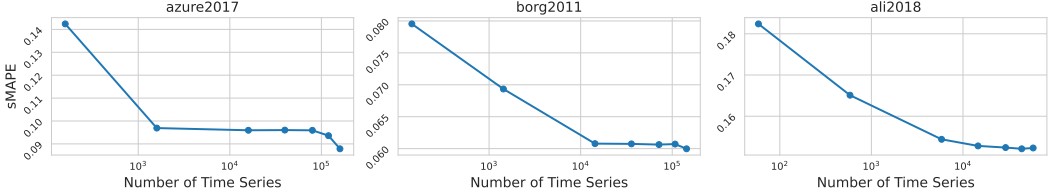

Figure 6: Performance curve against dataset size (number of time series pre-training collection). Model used here is the base size, trained for 100,000 iterations for `azure2017` and 25,000 iterations for `borg2011` and `ali2018`.

on qualitative analysis visualized in Appendix F, we observe that the model fails to accurately forecast patterns appearing in the data at a lower frequency, which is possibly one limitation of a global model.

### 4.4 SCALING

We perform further investigations into the scaling behavior of these models by pre-training a sequence of models for an increasing number of iterations (Figure 5), and increasing dataset size (Figure 6). On our largest dataset, `azure2017`, we observe a clear trend where performance improves as model size and number of observations increase. This relationship is more ambiguous on the smaller `borg2011` and `ali2018` datasets, but a more fine-grained plot of the validation performance on intermediate checkpoints in Appendix G shows that the models are overfitting on these datasets. This finding is supported by evidence that repeating samples during pre-training leads to improving pre-training loss but worsening downstream performance (Raffel et al., 2020). While we may not be repeating inputs in terms of number of observations (i.e. time series * time steps), inputs could be similar across time, or contain redundant samples. We further see that dataset size and diversity is critical from Figure 6.

## 5 CONCLUSION

In this work, we introduced three large-scale time series forecasting datasets in the CloudOps domain to fuel further research into pre-training of large time series models. We pave the way for future work in this area by introducing a promising candidate architecture after a series of experiments, showing that performance scales with increasing training length, dataset size, and model size. We establish a benchmark on these datasets with classical and deep learning baselines, showing that our proposed architecture is a strong pre-trained zero-shot forecaster.

**Limitations & Going Forward** Despite the extensive results we have presented, one limitation of this work is that our experiments are not fully comprehensive – while the ideal would be to perform a grid search on all possible combinations of design choices and hyperparameters, we are unable to do so due to computational constraints. Furthermore, with more resources, the ideal would be to establish a scaling law for time series models. Looking forward, pre-training on a cross-collection and even cross-domain pre-training dataset is at the top of our minds. While doing so may unlock powerful generalization capabilities, we anticipate a major challenge – the heterogeneity of time series data. This in itself subsumes many sub-challenges, including the problem of multiple frequencies (minutely, hourly, daily , etc. sampling rates), time series patterns at different scales, raising questions about how to handle input length to the model, and heterogeneous input space (different covariates).

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

# A CLOUDOPS DATASETS DETAILS

## A.1 DATASET ATTRIBUTES

Table 6: Summary of data attributes for the CloudOps datasets.

|  | azure2017 | borg2011 | ali2018 |
|---|---|---|---|
| **Target** | Avg. CPU utilization | CPU rate
Canonical memory usage | CPU utilization percent
Memory utilization percent |
| **Static Real** | Virtual core count
Memory
Deployment size | - | - |
| **Past Dynamic** | Min. CPU utililization
Max. CPU utilization | Assigned memory usage
Unmapped page cache
Total page cache
Local disk space usage
Sample portion | CPI
Mem GPS
MPKI
Net in
Net out
Disk I/O percent |
| **Dynamic** | Date/Time | Date/Time | Date/Time |

Table 6 summarizes the data attributes in the processed versions of the CloudOps datasets. Targets refer to the time series that we are interested in forecasting. Static real values covariates are real values (single value, not time series). Past dynamic covariates are time series features, of which we only have access to the context/lookback window, i.e. $Z_{1:t}$. Dynamic covariates are time series features, of which we have access to both the context/lookback window and the prediction range/horizon, i.e. $Z_{1:t+H}$. All values are reals, we do not consider any categorical covariates in this work.

## A.2 DATA COLLECTION

### A.2.1 AZURE VM TRACES 2017

The azure2017 dataset (Cortez et al., 2017) was downloaded from https://github.com/Azure/AzurePublicDataset. A user of Azure cloud can create one or more *subscriptions*, and a *deployment* is a set of VMs that the customer groups and manages together. For each observation, we have access to the *encrypted subscription id*, *encrypted deployment id*, and *encrypted VM id*. The original format is a row-based dataset, and the schema is as follows (only the columns used for the final processed dataset, with full details available in the link):

- Average CPU utilization
- Minimum CPU utilization
- Maximum CPU utilization

Each row represents an aggregate of 5 minutes of VM CPU utilization readings, thus the average/minimum/maximum CPU utilization readings are the statistic for that 5 minute window that the row represents.

**Data cleaning** We convert the row format data to columnar format, grouping observations by the unique VM id, and sorting the time series by timestamp. Thereafter, we performed data cleaning by performing the following steps in order:

1. Remove duplicate timestamps for each VM id
2. Fill missing values with nulls
3. Filter out time series with the following characteristics:
    - Too short – time series which are shorter than $48 * (12 + 1 + 1) = 672$ time steps. This was selected based on the test set being 12 non-overlapping windows of horizon length 48, 1 horizon for validation, and 1 additional horizon for training.

- Too many missing values – time series which have more than $0.125\%$ missing values
- Constant time series – uninformative time series which only have a single value across time

4. Adjust timestamps – the timestamps are anonymized, and record the relative time with the reference point 0 being the the time of the first record. We assume the reference point 0 to be 15 November 2016, 0:00:00, the middle of the month of the starting date of the original unanonymized dataset (not publicly available) (Cortez et al., 2017).

**Data split**  To avoid data leakage from the pre-training set to the train-test set, we ensure two things:

1. All time series in the train-test set are end-aligned to the final time stamp in the entire dataset. This means that the time stamps of the test region will never appear in the pre-training set (recall from Figure 2 that the test region is removed from the pre-training set), thus preventing temporal data leakage (Hewamalage et al., 2023).

2. The data split between pre-training set and train-test set is done at the top level relationship between time series, subscription id for `azure2017`. VM ids with different subscription ids will not be related in any way based on the available information of the original raw dataset. This avoids data leakage from the pre-training set to the train-test set since time series in the pre-training set are not related to those in the train-test set.

Based on these principles, we select approximately 10% of the entire dataset to be in the train-test set. This is done by randomly selecting $n = \text{round}(\text{n\_valid\_subscriptions} * \frac{10\% * \text{n\_time\_series}}{\text{n\_valid\_time\_series}})$ subscriptions from the set of valid subscriptions.

### A.2.2  Borg Cluster Data 2011

The `borg2011` dataset (Wilkes, 2011) was downloaded from https://github.com/google/cluster-data. Borg comprises a logically centralized cluster scheduler master, and a large number of machines (nodes), each of which runs a local management agent. Each such deployment is called a cell, and is operated as a single management unit. A *user* initiates a *job* at the cell, which comprises one or more *tasks*. We have access to the *user id*, *job id*, and *task id*. Measurements are made at the task level. Similar to `azure2017`, the original format is a row-based dataset, with the following columns being of interest.

- CPU rate
- Canonical memory usage
- Assigned memory usage
- Unmapped page cache
- Total page cache
- Local disk space usage
- Sample portion

Each row reports usage values from each measurement period of 5 minutes. Within each measurement period, measurements are typically taken at 1 second intervals. However, there are cases thus, *sample portion* refers to the ratio between number of expected samples to the number of observed samples. These measurements are then aggregated, providing the mean for each period. Further details of each measurement can be found in the original documentation at the above link.

**Data cleaning**  Data cleaning is performed in a similar manner to that of `azure2017`, where we instead filter time series which have more than 1% missing values, and set the reference point to 1 May 2011, 19:00:00.

**Data split**  The data split is performed in a similar manner to that of `azure2017`, with the top level attribute being User.

### A.2.3    ALIBABA CLUSTER TRACE 2018

The `ali2018` dataset (Guo et al., 2019) was downloaded from `https://github.com/alibaba/clusterdata`. The dataset is sampled from one of Alibaba's production clusters. In particular, we processed the trace of online services/long running applications. Measurements are at the *container* level, and containers with the same *App DU* belong to the same application group. We have access to the *container id* and *App DU id*. Similar to `azure2017` and `borg2011`, the original format is a row-based dataset, and below are the columns of interest.

- CPU util percent
- Mem util percent
- CPI
- Mem GPS
- MPKI
- Net in
- Net out
- Disk I/O percent

Unlike `azure2017` and `borg2011`, observations are irregularly sampled. Thus, we aggregate all metrics by splitting all observations into windows of 5 minute intervals, and report the average of that window.

**Data cleaning**    Data cleaning is performed in a similar manner to that of `azure2017` and `borg2011`, where we filter time series which have more than 1% missing values, and set the reference point to 1 January 2018, 12:00:00.

**Data split**    The data split is performed in a similar manner to that of `azure2017` and `borg2011`, with the top level attribute being App DU.

### A.3    DATA ANALYSIS

### A.3.1    DATA SPLIT

One concern regarding the in-collection setting lies in whether there is at all a difference in distribution between the pre-train and train-test sets. As discussed in Appendix A.2 our data splitting strategy is based on non-overlapping top level attributes, thus we expect the time series patterns/distributions to be different. We perform an analysis to verify this hypothesis and to highlight the challenge of the in-collection pre-training setting. Due to the size of the datasets, we perform the following analyses on time series belonging to a random subset of 100 top level attributes for the pre-train and train-test sets (i.e. we consider all time series belonging to 100 subscription ids for the pre-train set, and all the time series belonging to 100 subscription ids for the train-test set for `azure2017`. For all three datasets, we consider the "cpu utilization" time series.

**Qualitatively**, we can perform the simple analysis of visualizing the empirical distribution by plotting a histogram of the time series values. As observed in Figure 7, when we visualize the train and test regions of the train-test set separately, we see that the gap between the train and pre-train sets is larger than that of the train and tests set. This helps us verify that there is a distribution shift between the pre-train and train-test splits.

**Quantitatively**, we obtain a measure of distribution shift between the pre-train set and train-test set. First, we extract 12 representative features (Godahewa et al., 2021). These features include spectral entropy, strength of trend (hourly and daily), strength of seasonality (hourly and daily), first-order autocorrelation for the series, differenced series, and twice differenced series, the sum of squares of the first 10 autocorrelation coefficients in each case, and the optimal box-cox transformation parameter. Here, since we are dealing with large-scale datasets with potentially any seasonality pattern, we consider both hourly and daily seasonality by defining the frequency parameter to be $60/5$ and $24 * 60/5$ respectively. Each time series is now represented by a 7-dimensional feature vector. We use the Wasserstein distance (Flamary et al., 2021) to measure the distance between the

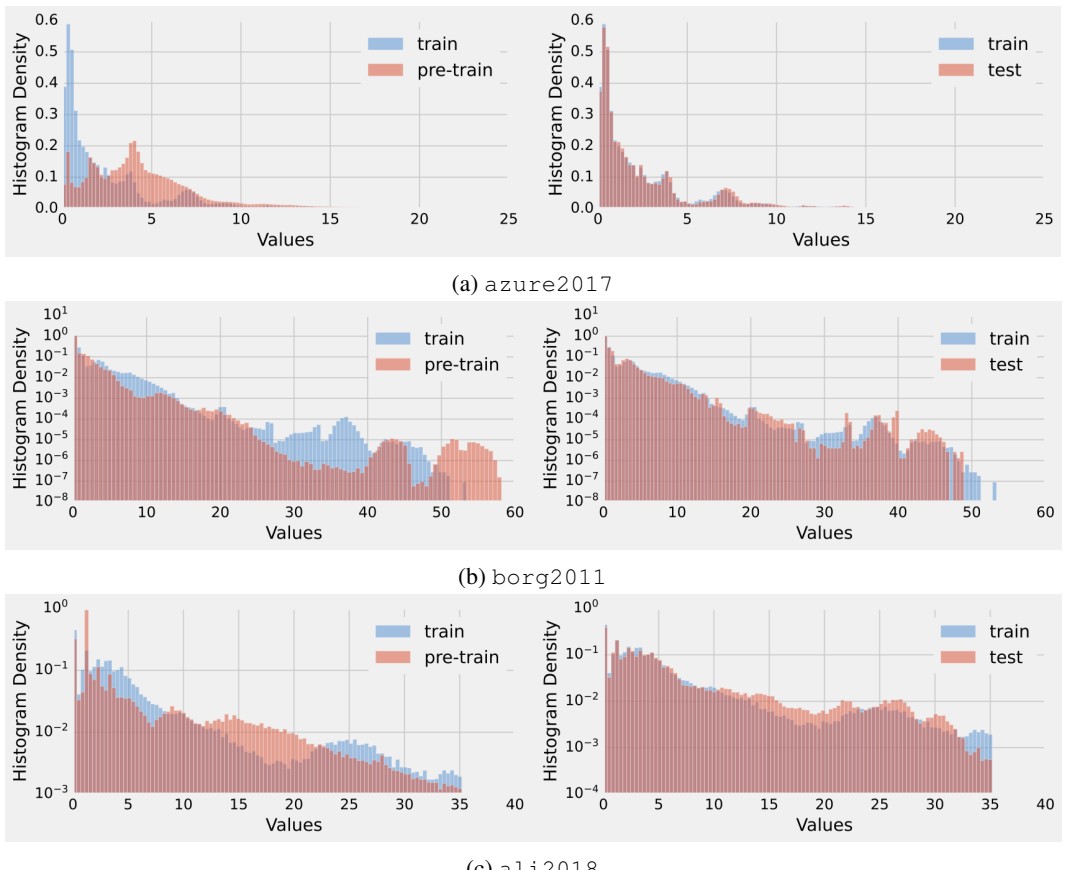

Figure 7: Left: Histogram plot of the train region of the train-test set compared to the pre-train set. Right: Histogram plot of the train and test regions of the train-test set, plotted separately. We remove anomalies for `azure2017` and `ali2018`, and report in a log scale for `borg2011` and `ali2018`.

Table 7: Wasserstein distance between two sets of data points. For rows 2 (Pre-train A vs Pre-train B) and 3 (Train-test A vs Train-test B), we randomly split the set into two equal, non-overlapping subsets, reporting the mean and standard deviation over 10 random splits.

| | azure2017 | borg2011 | ali2018 |
|---|---|---|---|
| Pre-train vs Train-test | 5.26 | 1.69 | 9.81 |
| Pre-train A vs Pre-train B | $0.160_{\pm 0.013}$ | $0.077_{\pm 0.024}$ | $0.361_{\pm 0.024}$ |
| Train-test A vs Train-test B | $0.802_{\pm 0.270}$ | $0.296_{\pm 0.098}$ | $0.545_{\pm 0.268}$ |

distributions of the pre-train and train-test sets. As observed in Table 7, the distance between pre-train and train-test set is much larger compared to the baseline of random subsets of the pre-train set with itself, and similarly for the train-test set. This again highlights the challenge of distribution shift present in the data split.

### A.3.2 DATASET DIVERSITY

We perform a similar analysis on the diversity of time series patterns/distributions across the three datasets. Similar to Appendix A.3.1, we perform the analysis on the "cpu utilization" time series. Figure 8 visualizes the first two principle components of the features after performing a Principle Component Analysis transform. Table 8 performs a similar quantitative analysis of the diversity between time series from the various datasets.

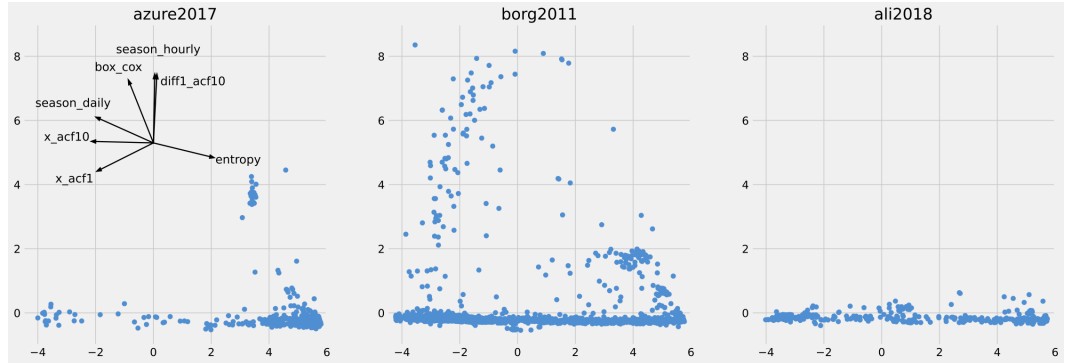

Figure 8: Scatter plots of the low-dimensional feature space generated by PCA across ACF1, ACF10, ACF10 of the differenced time series, seasonal strength (hourly and daily), entropy, and Box-Cox lambda over the three datasets. `azure2017` includes the directions of the various features, which are the same across all plots.

Table 8: Pairwise Wasserstein distance between all datasets. Diagonal is generated by randomly splitting each dataset into two equal, non-overlapping subsets, reporting the mean and standard deviation over 10 random splits.

|  | azure2017 | borg2011 | ali2018 |
|---|---|---|---|
| azure2017 | $0.802_{\pm0.270}$ | 42.86 | 20.85 |
| borg2011 | 42.86 | $0.296_{\pm0.098}$ | 10.77 |
| ali2018 | 20.85 | 10.77 | $0.545_{\pm0.268}$ |

## B   EVALUATION METRICS

**Symmetric Mean Absolute Percentage Error**   Percentage errors are unit-free, being normalized by the the absolute target values. We first define the error of a univariate time series to be,

$$\boldsymbol{e}_j^{(i)} = \boldsymbol{y}_j^{(i)} - \hat{\boldsymbol{y}}_j^{(i)}$$

where $\boldsymbol{y}_j^{(i)}$ and $\hat{\boldsymbol{y}}_j^{(i)}$ are the target and predicted values of $i$-th time series and $j$-th time step, respectively. Then, the sMAPE of the $i$-th time series is defined to be

$$\text{sMAPE} = \frac{200}{H} \sum_{j=t+1}^{t+H} \frac{|\boldsymbol{e}_j^{(i)}|}{|\boldsymbol{y}_j^{(i)}| + |\hat{\boldsymbol{y}}_j^{(i)}|}.$$

The sMAPE for multivariate datasets is simply the average over multivariate dimensions.

**Continuous Ranked Probability Score**   Before we can introduce the CRPS, we need to introduce the weighted quantile loss (Park et al., 2022), which is a metric normalized over the test set. We first define the $\alpha$-quantile loss, also known as the pinball loss at quantile level $\alpha$, to be:

$$\Lambda_\alpha(q, y) = (\alpha - \mathbf{1}_{\text{y<q}})(y - q).$$

The weighted quantile loss is then the normalized sum of quantile losses,

$$\text{wQL}[\alpha] = 2\frac{\sum_{(i,j)\in\Omega} \Lambda_\alpha(\hat{\boldsymbol{q}}_j^{(i)}(\alpha), \boldsymbol{y}_j^{(i)})}{\sum_{(i,j)\in\Omega} |\boldsymbol{y}_j^{(i)}|},$$

where $\Omega = \{(i,j) \in \mathbb{Z}^2 : 1 \leq i \leq n, \tau_i + 1 \leq j \leq T_i\}$.

The CRPS is a proper scoring rule (Matheson & Winkler, 1976; Gneiting & Raftery, 2007), meaning that it is minimized when the predictive distribution is equal to the distribution from which the data is drawn.

$$\text{CRPS} = \int_0^1 2\Lambda_\alpha(F^{-1}(\alpha), y)d\alpha$$

However, we are unable to evaluate this quantity since we generally are not able to compute the integral in closed form and only have access to a finite number of quantile predictions. The approximation of the CRPS is an average of the weighted quantile loss over $K$ quantiles, and thus is also known as the mean weighted quantile loss.

$$\text{CRPS} \approx \frac{1}{K} \sum_{k=1}^{K} \text{wQL}[\alpha_k]$$

**CRPS-sum**   The CRPS-sum metric was introduced in (Salinas et al., 2019) as an extension to the CRPS to evaluate multivariate probabilistic forecasts, and showed in de Bézenac et al. (2020) to be a proper scoring rule.

$$\text{CRPS}_{\text{sum}} = \text{CRPS}(F_{\text{sum}}, \sum_{k} \boldsymbol{y}_{j,k}^{(i)})$$

where $F_{\text{sum}}$ is the distribution of the sum of the multivariate dimensions.

## C   IMPLEMENTATION DETAILS

Experiments are implemented in PyTorch (Paszke et al., 2019) and ran on NVIDIA A100-40GB GPUs. For pre-training, we use TF32 precision with Distributed Data Parallel (DDP) on multiple 4 or 8 GPUs for pre-training and use gradient accumulation, depending on resource constraints.

### C.1   ARCHITECTURE DETAILS

We use a standard Transformer layer (Vaswani et al., 2017), with pre-LN modification (Xiong et al., 2020). Each layer comprises a multi-head self-attention block, a cross-attention block for decoder layers, followed by a feedforward block. The self-attention block has $n_{\text{head}} = 6$ heads, leading to each key/value dimension of each head being $d_{\text{kv}} = 64$. The feedforward block is a composition of a linear layer with output dimension of $d_{\text{ff}} = 1536$, followed by a GeLU non-linearity (Hendrycks & Gimpel, 2023), and another linear layer mapping back to $d_{\text{model}}$. The baseline probabilistic head is a Student-T distribution, with an independence assumption for multivariate datasets. Given the output representation from the Transformer, we learn projection layer, applying the appropriate non-linearity, to predict the parameters of the Student-T distribution (mean, variance, degrees-of-freedom) for each time step. Weight decay of $0.1$ is applied, with biases and LayerNorm parameters being omitted. Teacher forcing (Williams & Zipser, 1989) is applied at training for the encoder-decoder (IMS) architecture.

### C.2   FEATURES

**Normalization**   For all methods, we apply instance normalization on target values. Given a lookback window of length $L$, we calculate the mean and standard deviation, which is subsequently used to normalize input targets and unnormalize predictions. It is defined as

$$\hat{\boldsymbol{\mu}}_t^{(i)} = \frac{1}{L} \sum_{j=t-L+1}^{t} \boldsymbol{y}_j^{(i)}; \quad \hat{\boldsymbol{\sigma}}_t^{(i)} = \sqrt{\frac{1}{L} \sum_{j=t-L}^{t} (\boldsymbol{y}_j^{(i)} - \hat{\boldsymbol{\mu}}_t^{(i)})^2 + \varepsilon}$$

$$\text{norm}(\boldsymbol{y}_t^{(i)}) = \frac{\boldsymbol{y}_t^{(i)} - \hat{\boldsymbol{\mu}}_t^{(i)}}{\hat{\boldsymbol{\sigma}}_t^{(i)}}$$

$$\text{unnorm}(\hat{\boldsymbol{y}}_{t+h}^{(i)}) = \hat{\boldsymbol{\sigma}}_t^{(i)} * \hat{\boldsymbol{y}}_{t+h}^{(i)} + \hat{\boldsymbol{\mu}}_t^{(i)}, \forall h = 1, \ldots, H$$

where $\varepsilon$ is a small positive value.

**Log Scale**   We generate a static real feature, which is simply the log of the standard deviation, $\log(\hat{\boldsymbol{\sigma}}_t^{(i)})$, to impart knowledge of the normalization to the neural network models.

**Date/Time Features** For time series of 5min sampling frequency, we generate minute-of-hour, hour-of-day, day-of-week, day-of-month, day-of-year features. These are real valued features shifted to the range $[-0.5, 0.5]$.

$$\text{feature}(x) = \frac{x}{\text{max}_{\text{feature}}} - 0.5$$

For example, for minute-of-hour features, $x \in \{0, 1, \dots, 59\}$ and $\text{max}_{\text{feature}} = 59$.

**Lag Features** Lag features are lagged target values generated to enhance the information per time step, similar to time series shingling (Guha et al., 2016). The lag values are dependent on the sampling frequency, and we use the GluonTS implementation[1] with a maximum lag of 1200.

## D FURTHER ABLATIONS

### D.1 PROBABILISTIC HEADS

Probabilistic forecasting requires a predictive probability distribution rather than just a point forecast. One useful abstraction in deep probabilistic forecasting models is the idea of a probabilistic head, which is the layer responsible for mapping the representation produced by the Transformer, into the predictive distribution. In this section, we identify several simple probabilistic heads which can be easily plugged into this Transformer framework in a composable manner, and perform an empirical comparison.

**Parametric distributions** are the most straightforward approach to probabilistic forecasting, assuming some simple family of parametric distributions (Salinas et al., 2020). We select the Student-T distribution, predicting the location, scale, and degrees of freedom parameters. For multivariate datasets, we compare a Student-T distribution with an independence assumption, only predicting the diagonals of the scale matrix, and a full multivariate Student-T distribution, predicting the full scale matrix.

**Quantile functions** (QF) can be used to predict quantiles when the parametric form is unknown or when the full distribution is not required (Wen et al., 2017). Spline quantile functions (SQF) (Gasthaus et al., 2019), incremental quantile functions (IQF), and incremental spline quantile functions (ISQF) (Park et al., 2022) were introduced to solve various issues of QFs, such as quantile crossing and inter/extrapolation to quantiles not available at train time. We extend these quantile methods to the multivariate setting by naively predicting separate QFs for each dimension.

**Normalizing flows** allows us to learn more complex and flexible distributions where density evaluation and sampling can be computed efficiently. Rasul et al. (2021b) introduces two variants of conditional flow modules as probabilistic forecasting heads in the multivariate setting, RealNVP and Masked Autoregressive Flows (MAF).

Table 9: Results of probabilistic heads on the masked encoder Transformer variant.

| | | azure2017 | | borg2011 | | ali2018 | |
|---|---|---|---|---|---|---|---|
| | | sMAPE | CRPS | sMAPE | $\text{CRPS}_{\text{sum}}$ | sMAPE | $\text{CRPS}_{\text{sum}}$ |
| Student-T | Independent | **0.094** | **0.093** | **0.052** | 0.115 | **0.149** | 0.016 |
| | Multivariate | - | - | 0.053 | 0.119 | 0.150 | **0.011** |
| Quantile Function | QF | 0.136 | 0.118 | 0.058 | 0.117 | 0.152 | 0.017 |
| | IQF | 0.141 | 0.124 | 0.066 | 0.119 | 0.152 | 0.016 |
| | SQF | 0.135 | 0.117 | 0.060 | 0.116 | 0.152 | 0.017 |
| | ISQF | 0.137 | 0.119 | 0.058 | **0.113** | 0.152 | 0.016 |
| Normalizing Flow | RealNVP | - | - | 0.494 | 0.180 | 0.267 | 0.077 |
| | MAF | - | - | 0.510 | 0.157 | 0.296 | 0.081 |

**Results** Overall, the independent Student-T distribution proves to be a a simple and robust choice, outperforming quantile functions and normalizing flows. We note that while normalizing flows were originally proposed for high-dimensional forecasting problems, our datasets have low dimensionality. Furthermore, the addition of a flow neural network leads to further optimization issues with more hyperparameters to tune, leading to severe underperformance in our experiments.

---

[1]https://github.com/awslabs/gluonts/blob/dev/src/gluonts/time_feature/lag.py

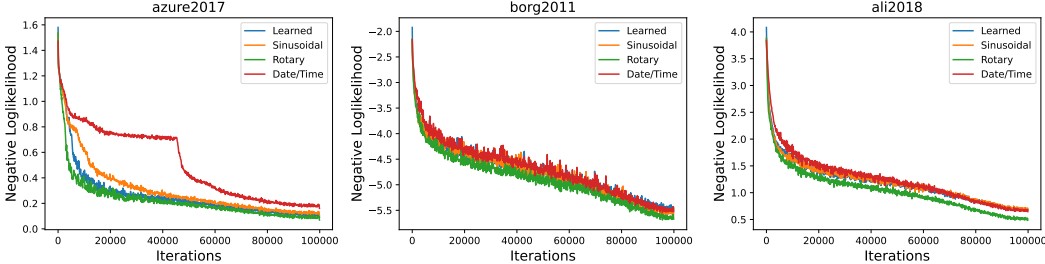

Figure 9: Pre-training loss for various positional encoding methods. Learned/Sinusoidal/Rotary are without date/time features.

## D.2 POSITIONAL ENCODINGS

Transformers rely on the attention mechanism to process temporal relationships between representations across time steps. One issue of the attention mechanism is that it is permutation equivariant, and requires positional encodings to encode positional information. A natural approach to encoding positional information in time series is to leverage date/time features. These include features such as the minute-of-hour, day-of-week, etc., depending on the sampling frequency. In this section, we study the impact of leveraging these features to encoding positional information, versus widely used approaches in the Transformer literature.

**Sinusoidal Positional Encodings (SPE)** (Vaswani et al., 2017) are absolute positional encodings, generated through a predefined sinusoidal function, and added to the representations before being fed into the Transformer.

**Learned Positional Embeddings** (Devlin et al., 2019) are learnable absolute positional encodings. Similar to SPEs, they are added to the representations before being fed into the Transformer, but rather than generated through a predefined function, they are randomly initalized and learned via gradient descent during training.

**Rotary Positional Embeddings (RoPE)** (Su et al., 2021) encodes the absolute position with a rotation matrix, and the explicit relative position dependency in the self-attention formulation. RoPE has been rapidly adopted as the positional encoding of choice by many recent LLMs (Chowdhery et al., 2022; Black et al., 2022; Nijkamp et al., 2023).

Table 10: Results of positional encodings on the masked encoder Transformer variant.

|  | azure2017 | | borg2011 | | ali2018 | |
|---|---|---|---|---|---|---|
|  | sMAPE | CRPS | sMAPE | CRPS$_{sum}$ | sMAPE | CRPS$_{sum}$ |
| Date/Time | 0.094 | 0.093 | 0.052 | 0.115 | 0.149 | 0.016 |
| SPE | 0.090 | 0.088 | 0.051 | 0.114 | 0.149 | 0.016 |
| SPE +Date/Time | 0.089 | 0.088 | 0.051 | 0.115 | 0.148 | 0.016 |
| LPE | 0.089 | 0.087 | 0.051 | **0.113** | 0.149 | 0.016 |
| LPE +Date/Time | 0.090 | 0.088 | **0.050** | 0.114 | 0.149 | 0.016 |
| RoPE | **0.088** | **0.086** | 0.052 | 0.116 | 0.149 | 0.016 |
| RoPE +Date/Time | **0.088** | **0.086** | 0.051 | 0.114 | 0.149 | 0.017 |

**Results** Results of various positional encoding methods with and without date/time features are presented in Table 10. Pre-training loss curves are visualized in Figure 9. Notably, date/time information are not critical features for forecasting, achieving sub-optimal pre-training and can be removed without harming performance. All three positional encodings yield significant improvements especially in `azure2017` and `borg2011`, whereas there is little to no impact on `ali2018`. RoPE achieves the best pre-train loss across all datasets, and adding date/time features in conjunction with the positional encodings yield no net negative effects, thus we opt to use RoPE + date/time features in subsequent experiments.

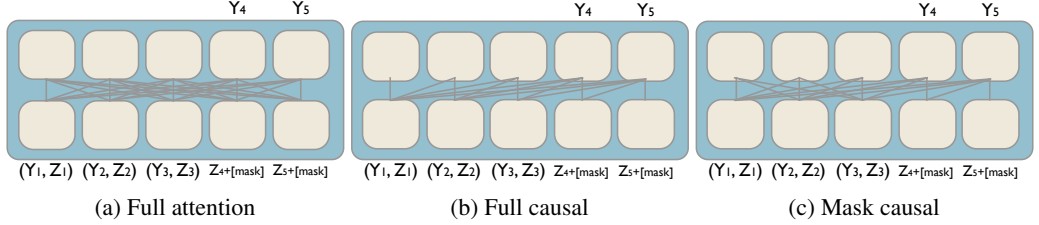

Figure 10: Attention mask schemes for the masked encoder architecture. (a) Full attention applies bidirectional encoding across all inputs. (b) Full causal applies unidirectional decoding across all inputs. (c) Mask causal applies bidirectional encoding across the context window, and unidirectional decoding across masked inputs.

### D.3 ATTENTION MASKS

While the masked encoder introduced in BERT (Devlin et al., 2019) was used in pre-training, masked reconstruction was not used in downstream tasks which mainly focused on obtaining a representation of the entire input. Thus, they focused on the bidirectional encoder architecture, and did not consider other attention masking schemes.

Causal attention masks can be used to differentiate between encoding and decoding, i.e. full attention for encoding and causal attention for decoding. Dong et al. (2019) introduced various attention masking strategies for a unified Transformer architecture in the context of NLP. While the various masking strategies correspond to different downstream tasks in natural language processing (e.g. full attention/bidirectional encoding for extractive question answering and full causal/unidirectional decoding for long text generation), it is unclear which paradigm time series forecasting fits in. On the one hand, we could argue that past time steps should not attend to future time steps, on the other hand, attending to future time steps could help in extracting seasonal information for example. Figure 10 illustrates the various attention masking schemes for the masked encoder architecture.

Table 11: Results of masking strategies.

| | azure2017 | | borg2011 | | ali2018 | |
|---|---|---|---|---|---|---|
| | sMAPE | CRPS | sMAPE | CRPS$_{sum}$ | sMAPE | CRPS$_{sum}$ |
| Full attention | 0.088 | 0.086 | 0.051 | 0.114 | 0.149 | 0.017 |
| Full causal | 0.088 | 0.086 | 0.051 | **0.111** | 0.148 | 0.016 |
| Mask causal | 0.088 | 0.086 | 0.050 | 0.113 | 0.148 | 0.016 |

**Results** We observe that attention masking plays a very minor role in performance in the `azure2017` and `ali2018`, with full and mask causal bringing some minor gains in `borg2011`. Overall, we consider these gains to be marginal and consider attention masking to play no major role for masked encoders for time series forecasting.

# E BASELINES

## E.1 CLASSICAL BASELINES

**Naive**   The naive forecast (Hyndman & Athanasopoulos, 2018) considers the last recorded value to be the forecast. We use the StatsForecast implementation (Garza et al., 2022) which also provides prediction intervals based on residuals and assuming a Gaussian distribution.

**AutoARIMA/ETS/Theta**   We use automatic versions of ARIMA, ETS, and Theta (Hyndman & Khandakar, 2008) implemented in the StatsForecast package (Garza et al., 2022). We fallback to the naive forecast when the model fails to produce a prediction.

**VAR**   We use the statsmodels implementation (Seabold & Perktold, 2010), performing lag selection based on the Akaike Information Criterion with the default maximum lags defined to be $12 * (nobs/100)^{1/4}$. We fallback to the naive forecast when faced with anomalous forecasts due to non-stationarity of the data. This is applied when the sum of absolute errors exceed the sum of labels.

## E.2 DEEP LEARNING BASELINES

For deep learning baselines, we perform hyperparameter tuning using Optuna (Akiba et al., 2019) across 15 runs for each model. The hyperparameter search range for model specific hyperparameters is given in Table 12, and defined as such by picking values surrounding the default values provided by their respective papers/official implementations. We also tune the learning rate in the range (1e-5, 1e-1), and (1e-8, 1e-2) for weight decay. A multiplicative learning rate scheduler with a factor of 0.5 is applied, reducing the learning rate every 3 consecutive epochs when validation loss does not decrease. Models are trained over 10,000 iterations with a batch size of 128. We perform early stopping after 1000 iterations based on validation loss aggregated and reported every 100 iterations. The best model is picked based on validation loss, and retrained on the whole training range based on those hyperparameters for the recorded number of epochs to ensure the model is trained on the full dataset. Methods which use parametric distribution output heads are defaulted to Student-T distribution, and methods originally proposed as point forecast models are modified to have Student-T distribution output heads unless otherwise specified. Below are further details for specific methods which special considerations are required.

**N-BEATS**   (Oreshkin et al., 2020) N-BEATS achieved state-of-the-art performance using an ensemble of 180 models, each being a large ResNet style neural network. Due to resource limitations, we train an ensemble of 18 models (showed in Oreshkin et al. (2020) to already achieve extremely strong performance) using hyperparameters specified in the original paper and implementation. These 18 models are the cartesian product of hyperparameters in Table 12. We use a learning rate of 1e-3 with a patience of 10.

**Auto/FEDformer**   (Wu et al., 2021; Zhou et al., 2022) As per previous approaches, Autoformer and FEDformer are implemented similarly with the same forecasting pipeline (e.g. date/time features and past dynamic covariates, lag features). For the output head, the architecture structure of Auto/FEDformer are not amenable to attaching a parametric distribution head due to the separate trend and seasonality components forming the forecast, rather than a representation which can be used to project into the distribution parameters. Thus, the forecast is taken to be a degenerate distribution to compute the CRPS metric.

**LinearFamily**   Zeng et al. (2023) introduces three variants of linear models, Linear, DLinear, and NLinear, which we call the Linear Family. These models were introduced as point forecast models. Since they directly map the lookback window into the forecast horizon without a hidden state, they are not amenable to having a probabilistic output head. We optimize the mean absolute error for this method.

**DeepTime**   (Woo et al., 2023) DeepTime introduces an efficient meta-optimization formulation, solving a ridge regression problem in the inner loop. This approach is not straightforward to extend to arbitrary output heads and loss functions, thus we also optimize the mean absolute error for this method.

Table 12: Hyperparameter search range for deep learning baselines.

|  | Hyperparameter | Values |
| --- | --- | --- |
| MQ-CNN (Wen et al., 2017) | num layers | 3 |
|  | channels | $\{20, 25, 30, 35, 40\}$ |
|  | kernel size | $\{[3, 3, 2], [7, 3, 3], [14, 7, 3]\}$ |
| N-BEATS | loss | $\{\text{smape, mase, mape}\}$ |
|  | model type | $\{\text{generic, interpretable}\}$ |
|  | context length multiplier | $\{7, 9, 11\}$ |
| TFT (Lim et al., 2021) | num heads | $\{2, 4, 8\}$ |
|  | hidden size | $\{16, 32, 64\}$ |
| Autoformer (Wu et al., 2021) | factor | $\{2, 3, 4, 5\}$ |
|  | moving avg | $\{13, 25, 37\}$ |
|  | d_model | 512 |
|  | num heads | 8 |
|  | num encoder layers | $\{1, 2, 3\}$ |
|  | num decoder layers | $\{1, 2, 3\}$ |
|  | dim_feedforward | 2048 |
| FEDformer (Zhou et al., 2022) | version | $\{\text{Fourier, Wavelets}\}$ |
|  | modes | 64 |
|  | mode_select | random |
|  | base | legendre |
|  | cross_activation | tanh |
|  | L | 3 |
|  | moving_avg | 24 |
|  | n_heads | 8 |
|  | d_model | 512 |
|  | num_encoder_layers | $\{1, 2\}$ |
|  | num_decoder_layers | $\{1, 2\}$ |
|  | dim_feedforward | 2048 |
| NSTransformer (Liu et al., 2022) | p_hidden_dims | $\{64, 128, 256\}$ |
|  | p_hidden_layers | 2 |
|  | num_encoder_layers | $\{1, 2, 3\}$ |
|  | num_decoder_layers | $\{1, 2, 3\}$ |
| PatchTST (Nie et al., 2023) | d_model | $\{128, 256, 512\}$ |
|  | num_layers | $\{2, 3, 4\}$ |
| LinearFamily (Zeng et al., 2023) | model type | $\{\text{Linear, DLinear, NLinear}\}$ |
|  | individual | $\{\text{True, False}\}$ |
| DeepTime (Woo et al., 2023) | d_model | $\{256, 512, 1024\}$ |
|  | num_layers | $\{3, 5, 7, 9\}$ |
| DeepAR (Salinas et al., 2020) | num layers | $\{1, 2, 3, 4\}$ |
|  | hidden size | $\{20, 25, \dots 80\}$ |
| TimeGrad (Rasul et al., 2021a) | num layers | $\{1, 2, 3, 4\}$ |
|  | hidden size | $\{20, 25, \dots 80\}$ |

Table 13: Hyperparameter details and corresponding sizes for pre-trained DeepAR and Autoformer models. We include details of "Ours-Base" from Table 4 for comparison.

|                 | Layers | $d_{\mathrm{model}}$ | $d_{\mathrm{ff}}$ | # heads | $d_{\mathrm{kv}}$ | # params |
|-----------------|--------|-------|-------|---------|-------|----------|
| DeepAR-Base     | 6      | 384   | -     | -       | -     | 6.6m     |
| Autoformer-Base | 6      | 384   | 1536  | 6       | 64    | 24.8m    |
| Ours-Base       | 6      | 384   | 1536  | 6       | 64    | 10.7m    |

### E.3 PRE-TRAINED BASELINES

**TS2Vec/CoST**  (Yue et al., 2022; Woo et al., 2022a) TS2Vec and CoST are self-supervised methods, originally pre-trained on the same data as the downstream task, due to dataset limitations. We train these models on the pre-training set for 100,000 iterations with a batch size of 512, with the default hyperparameters recommended in the papers. We then fine-tune a linear predictor on the train-test set using the same protocol as per Section 4.1.

**One Fits All**  (Zhou et al., 2023) OFA introduces the idea of fine-tuning an LLM pre-trained on text data for time series tasks. We fine-tune a GPT2 model using the same protocol as Section 4.1.

**Meta N-BEATS**  (Oreshkin et al., 2021) Meta N-BEATS is similar to N-BEATS, except that it is trained on the pre-train set over 100,000 iterations with a batch size of 512.

**(DeepAR/Autoformer)-Base**  We scale DeepAR and Autoformer models to a similar size as "Ours-Base", based on hyperparameters as detailed in Table 13. These models are then pre-trained on the pre-training set with the same methodology outlined in Section 4.1, over 100,000 iterations.

# F    FORECAST VISUALIZATIONS

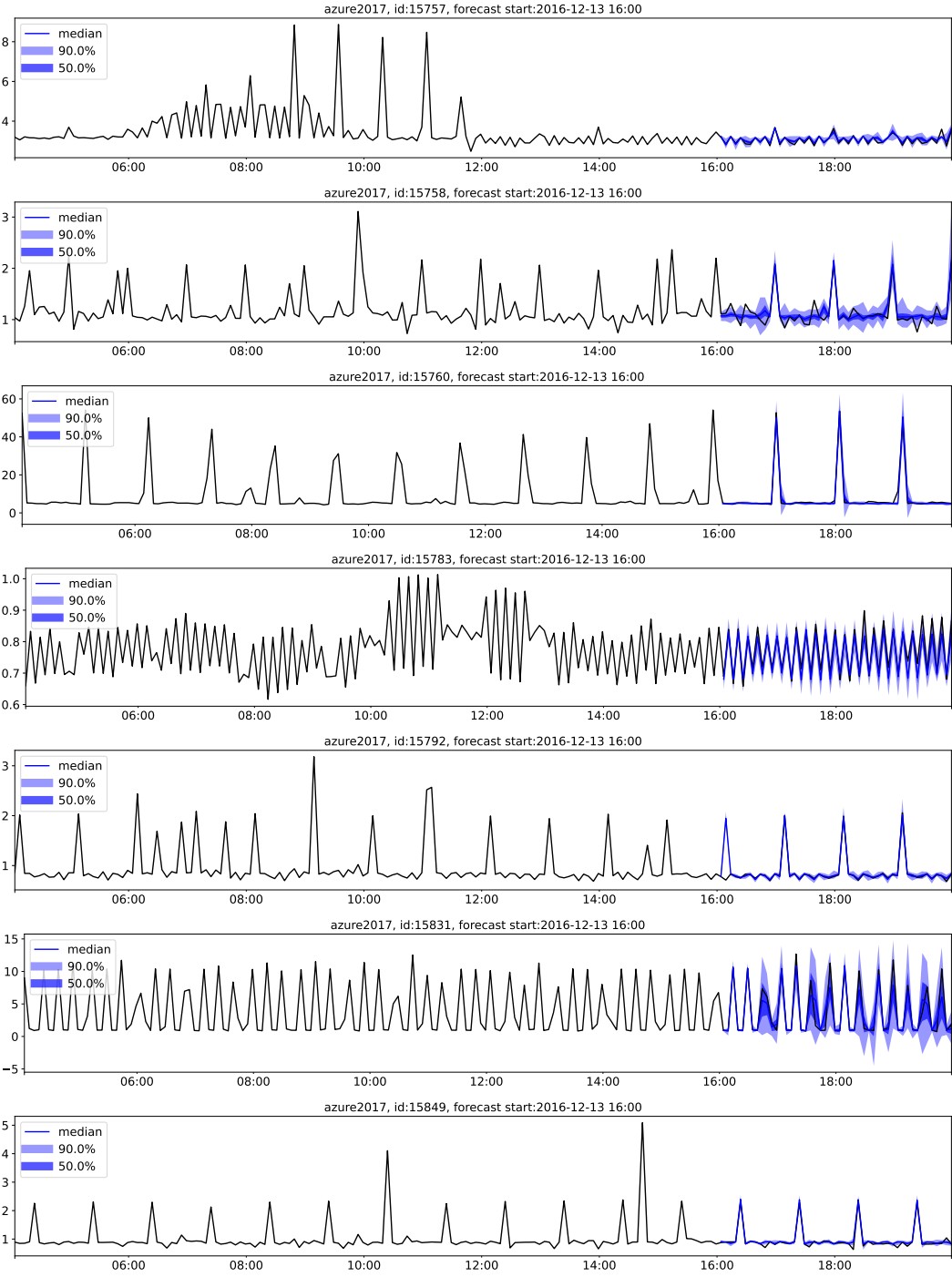

Figure 11: Visualizations of xLarge on `azure2017`.

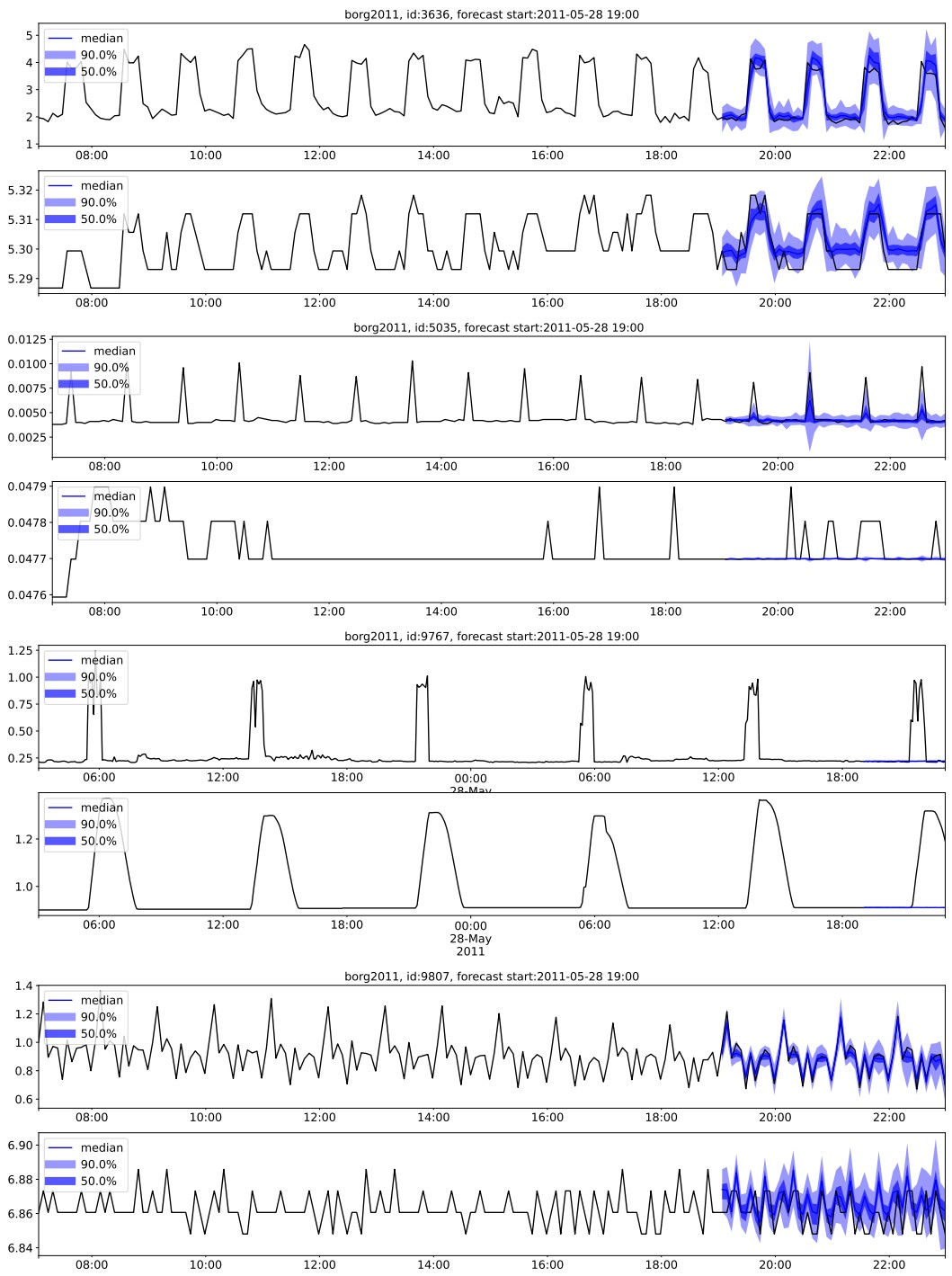

Figure 12: Visualizations of xLarge on `borg2011`. For each figure, the top plot represents CPU rate and the bottom plot represents canonical memory usage. The model manages to capture higher frequency patterns, but fails to forecast obvious patterns in id:9767 which occur at lower frequency.

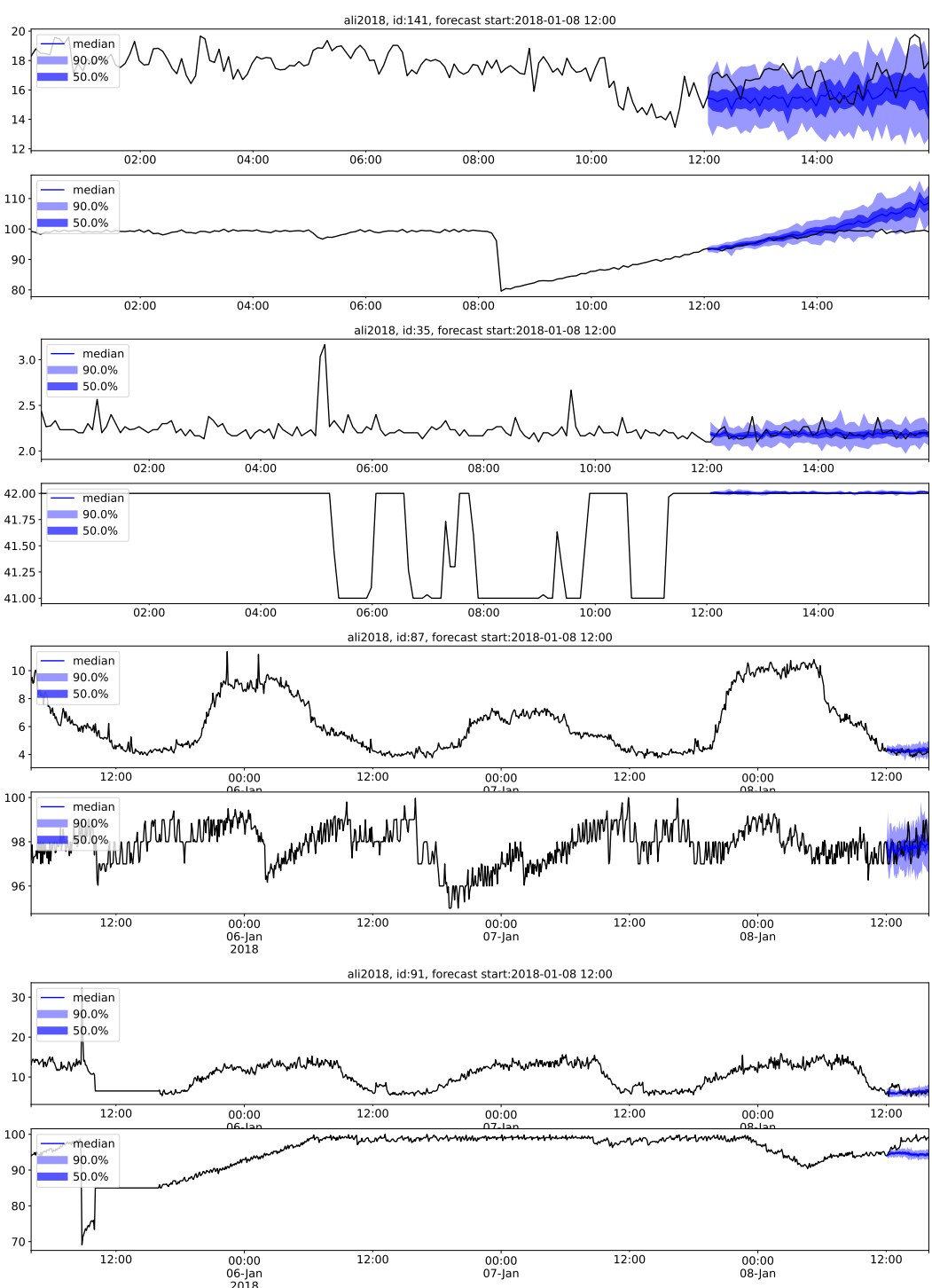

Figure 13: Visualizations of xLarge on ali2018. For each figure, the top plot represents CPU utilization percent, and the bottom plot represents memory utilization percent. We visualize a longer context window for this dataset to highlight the longer scale patterns.

# G  FINE-GRAINED SCALING PLOTS

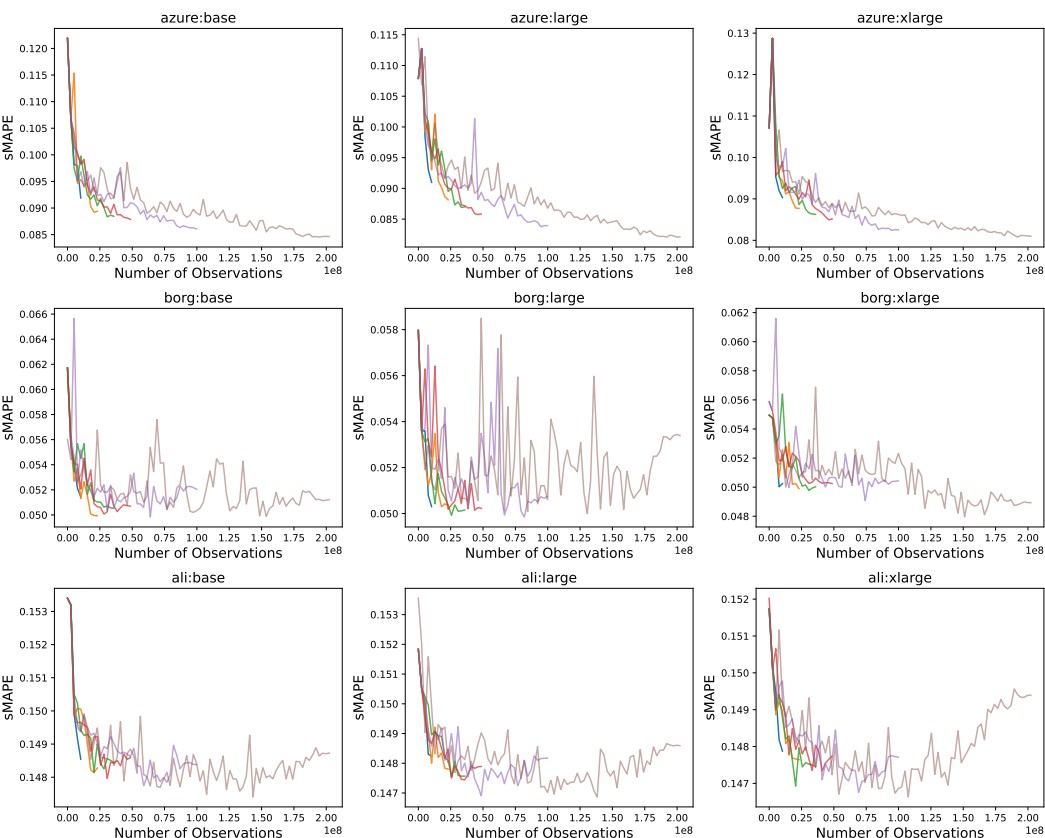

Figure 14: Fine-grained plot of validation error. Each curve represents the validation error across the training process of a particular model size trained for a particular number of iterations. The error is reported every 5000 iterations, i.e. we save a checkpoint every 5000 iterations during training, which is then used to report the validation error.

