# OpenReview forum: "Pushing the Limits of Pre-training for Time Series Forecasting in the CloudOps Domain"
_ICLR.cc/2024/Conference — Submitted to ICLR 2024_

### Official Review · Reviewer_boWu · 2023-10-30

**Soundness:** 3 good
**Presentation:** 3 good
**Contribution:** 2 fair
**Rating:** 5
**Confidence:** 3

**Summary:**

This paper introduces three new large datasets from the CloudOps domains with billions of observations for time series forecasting. Further, the authors provide a performance analysis including time series forecasting models for the cases where models are trained from scratch, fine tuned, and zero-shot. The authors show further evaluate different transformer variants and show that the proposed baseline achieves a 27% error reduction on largest dataset.

**Strengths:**

1. The authors provide three new large datasets from the CloudOps field. This contribution is relevant in the field of time series forecasting, as there is certain lack of this kind of datasets, holding back the progress in the topic of pretrained models or LLM-based forecasting models.
2. The authors provide an interesting evaluation of several existing models. These evaluations provide an interesting reference on how well-established models perform in these datasets, and how they compare with pretrained models.
3. The authors provide an interesting analysis on variants of transformer models, providing further insights on what approaches are more promising in the future.
4. The paper is well written and it makes an effort on having well structured terminology in these new and developing field.

**Weaknesses:**

The main weakness of the paper is that the contribution of this paper is basically three new datasets. I acknowledge the non-trivial effort that it implies to gather this kind of large scale datasets. Nevertheless, I share as well my concern that there is not much of an analysis of what are the main challenges that these datasets pose. For instance: 1/ Do they have missing values? 2/ how strong is the seasonality in these datasets? 3/ Is there any interesting trend in the data? Are there any distribution shifts (for instance, something like a black-friday regime, a change around covid lockdowns, etc)?

**Questions:**

1. what is the main challenge that these datasets pose?
2. how diverse are these three datasets? I understand that they come from the CloudOps field, but is there any fundamental difference between them?
3. Is there any distribution shift captured in these datasets?
4. How do these datasets look in the test windows used for evaluation? And how do the forecasts per model look? I understand that for ali2018 the forecasts are potentially similar as Naive performs very good, but what about azure 2017 where the proposed baselines are performing exceptionally well?

---

> ### Author Response · Authors · 2023-11-17
> **Response to Reviewer boWu (1/2)**
>
> Thank you for taking the time and effort to review our work. We hope that our responses below fully address your concerns and queries, and look forward to further discussion.
>
> __Q: The main weakness of the paper is that the contribution of this paper is basically three new datasets.__
>
> A: Please see our response to all reviewers, as this is a recurring issue raised.
>
> __Q: What is the main challenge that these datasets pose?__
>
> A: The main challenge of these datasets is the size. Due to the large number of time series in these dataset, we cannot make sweeping claims about the characteristics of time series in each dataset. One example is that these time series contain seasonality, as observed in Figures 10, 11, and 12 in Appendix F. However, the time series in one dataset can have differing seasonality periods, and some may not even have seasonality, or have cyclical patterns instead (repeating patterns but no fixed period). Unfortunately, these datasets are collected over a relatively short period of time (longest being 30 days), and for some datasets, there has been some obfuscation, whereby the actual date/time information is not given, and thus cannot be correlated back to real world events. These datasets indeed have missing values -- highlighted in Appendix A.2, we performed data cleaning, filtering from the raw data the time series with too many missing values.
>
> __Q: Is there a distribution shift captured in these datasets?__
>
> A: Thank you for raising this valuable question. We have performed further analysis to measure distribution shift across (1) pre-train and train-test sets, and (2) across time. In our updated Appendix A.3.1, we observe in Figure 7 that there is little temporal distribution shift across the train to test regions. This is expected, since the collection period of this dataset is over the period of 1 month, where there will not be significant shift over time. However, we do observe that there is a significant distribution shift across different time series in the pre-train set to train-test set.

---

> ### Author Response · Authors · 2023-11-17
> **Response to Reviewer boWu (2/2)**
>
> __Q: How diverse are these three datasets? I understand that they come from the CloudOps field, but is there any fundamental difference between them?__
>
> A: These datasets are relatively different. From a fundamental perspective, azure2017 is generated by first-party internal services of the Azure cloud system, whereas borg2011 has both non-production tasks (e.g. development, non-business-critical analyses) as well as revenue-generating user requests. ali2018 on the other hand is from a production cluster of long running applications.
>
> On the level of data attributes, as seen in Table 6 in Appendix A.1, azure2017 measures 1 target variable (CPU utilization) whereas borg2011 and ali2018 measures 2 target variables (CPU utilization and memory usage). Furthermore, each dataset measures unique auxiliary covariates, for example, azure2017 has unique static real features such as virtual core count, memory size, and deployment size.
>
> Next, we can do a qualitative comparison based on time series visualizations in Appendix F. azure2017 has high frequency cyclical patterns, whereas borg2011 has slightly lower frequency patterns. On the other hand, ali2018 has very low frequency seasonality, and contains significantly more trend information.
>
> Finally, we can perform a quantitative evaluation of the diversity of the datasets. Following [1], we can extract features for each time series -- the spectral entropy, strength of seasonality, strength of trend, first-order autocorrelation, and the optimal box-cox transformation parameter. Using these features, we measure the distance between the distribution of datasets using the Wasserstein distance (we have included this as Appendix A.3.2, including a scatter plot of the features in low-dimensional feature space generated by PCA).
>
> |               | azure2017    | borg2011      | ali2018       |
> | ------------- | ------------ | ------------- | ------------- |
> | azure2017 | 0.802 (0.27) | 42.86         | 20.85         |
> | borg2011      | 42.86        | 0.296 (0.098) | 10.77         |
> | ali2018       | 20.85        | 10.77         | 0.545 (0.268) |
>
>
> [1] Godahewa, Rakshitha, et al. "Monash time series forecasting archive." arXiv preprint arXiv:2105.06643 (2021).
>
> __Q: How do these datasets look in the test windows used for evaluation? And how do the forecasts per model look? I understand that for ali2018 the forecasts are potentially similar as Naive performs very good, but what about azure 2017 where the proposed baselines are performing exceptionally well?__
>
> A: Please see Appendix F for forecast visualizations across all three datasets in the test region. We observe that ali2018 is more challenging dataset, since based on qualitative observations, it has more noise and has long range seasonality patterns, compared to azure2017 which have fairly simple seasonality patterns.

---

> ### Author Response · Authors · 2023-11-20
>
> Dear Reviewer boWu,
>
> Thank you for taking the time and effort in providing the valuable review of our work thus far. We hope that our rebuttal has managed to provide clarity and address any remaining concerns you may have had on our work, and that you have had a chance to review our rebuttal. We are eager to address any remaining concerns, or if our rebuttal has addressed all your concerns, we hope that you could kindly leave an update to reflect this. Thank you so much for your time and effort.

---

> ### Author Response · Authors · 2023-11-21
>
> Dear Reviewer boWu,
>
> Thank you for taking the time and effort in providing a valuable review of our work thus far. As the discussion period is soon coming to a close, we hope that you have had the chance to review our rebuttal. We believe that our rebuttal has provided additional clarity and highlighted the strengths and contributions of our work. If our rebuttal has addressed your concerns, we hope that you could kindly leave an update to reflect this, and are more than willing to engage in further discussion if needed.

---

> > ### Comment · Reviewer_boWu · 2023-11-22
> >
> > I would like to thank the authors for their replies. I will keep my current score.

---

### Official Review · Reviewer_QeLY · 2023-10-31

**Soundness:** 2 fair
**Presentation:** 2 fair
**Contribution:** 2 fair
**Rating:** 5
**Confidence:** 4

**Summary:**

This paper presents pre-training time-series model in the cloud operations domain to enhance downstream forecasting accuracy. The authors conduct experiments to compare various model architectures and investigate the scaling laws impacting both model and data size. Their findings indicate promising results in zero-shot scenarios.

**Strengths:**

1. The paper introduces the first pre-trained time-series model specifically for cloud operation domains.
2. The study evaluates different model architectures in the context of pre-training and examines the effects of model and data size on performance with scaling laws.

**Weaknesses:**

1. The pre-training and zero-shot testing appear to be conducted within the same dataset. If this is correct, it raises concerns about the true generality of the zero-shot performance, particularly regarding its effectiveness in diverse or new datasets and domains.
2. The paper mainly focuses on benchmarking existing pre-training model architectures without significant novel adaptations or designs tailored to the specific requirements of the cloud operation domain.
3. Is the model randomly initialized? Given the effectiveness of the baseline one-fits-all model, an exploration into using pre-trained language models as initialization might be helpful.
4. It remains unclear whether deep learning benchmarks such as N-BEATS, Autoformer, and FEDformer are only trained on designated training sets. Exploring whether similar performance improvements could be achieved by training these models on the pre-training set, using the same loss function, might offer deeper insights. This also helps determine the source of the proposed model's performance gains—whether from the architecture itself or mostly from expanded datasets.

**Questions:**

See Weakness above.

---

> ### Author Response · Authors · 2023-11-17
> **Response to Reviewer QeLY (1/2)**
>
> Thank you for taking the time and effort to review our work. We hope that our responses below fully address your concerns and queries, and look forward to further discussion.
>
> __Q: The pre-training and zero-shot testing appear to be conducted within the same dataset. If this is correct, it raises concerns about the true generality of the zero-shot performance, particularly regarding its effectiveness in diverse or new datasets and domains.__
>
> A: The pre-training and zero-shot testing follows the "in-collection pre-training" setting as explained in Section 1 and Figure 2. This means that the pre-training set and test set comes from the same __collection__ of time series, but consist of different, non-overlapping time series. Detailed in Appendix A.2, we ensure that the data split between the pre-training and train-test sets is performed based on a *top level attribute*, ensuring that there is no overlap between them. Thus, we expect the time series patterns/distribution across pre-train and train-test sets to be different, making the in-collection pre-training setting to be meaningful and challenging.
>
> We verify this hypothesis in Appendix A.3 of our updated manuscript, where we perform further data analysis, qualitatively, and quantitatively, demonstrating the distribution shift between pre-train and train-test sets.
>
> 1. Qualitatively, we show in Figure 7, based on histogram plots comparing the (a) train region vs pre-train set, and (b) train region vs train-test region, that the gap in (a) is larger than the gap in (b).
> 2. Quantitatively, we obtain a measure of distribution shift between the pre-train set and train-test set. We do so by first encoding the time series into a feature vector, comprising of 7 representative features as suggested by [1] -- spectral entropy, strength of trend (hourly and daily), strength of seasonality (hourly and daily), first-order autocorrelation, and the optimal box-cox transformation parameter. Then, we use the Wasserstein distance to measure the distribution difference between pre-train and train-test distributions. Table 7 demonstrates, by comparing with a a baseline of the Wasserstein distance between two random subsets of the pre-train set, that there is indeed a difference in distribution between the pre-train and train-test sets.
>
> |                         | azure2017     | borg2011      | ali2018       |
> | ----------------------- | ------------- | ------------- | ------------- |
> | Pre-train vs Train-test | 5.26          | 1.69          | 9.81          |
> | Pre-train               | 0.160 (0.013) | 0.077 (0.024) | 0.361 (0.024) |
> | Train-test              | 0.802 (0.27)  | 0.296 (0.098) | 0.545 (0.268) |
>
> __Q: The paper mainly focuses on benchmarking existing pre-training model architectures without significant novel adaptations or designs tailored to the specific requirements of the cloud operation domain.__
>
> A: We address this in our response to all reviewers since this is a recurring issue amongst reviewers.
>
> __Q: Is the model randomly initialized? Given the effectiveness of the baseline one-fits-all model, an exploration into using pre-trained language models as initialization might be helpful.__
>
> A: Yes, the model is randomly initialized for pre-training. The direction of adapting pre-trained weights from other data modalities such as language is indeed a fascinating emerging area of research. It is an alternative to pre-training on time series data when large-scale datasets are not available. Thus, while it is outside the scope of our work, we have included OneFitsAll as a related work and relevant baseline.

---

> ### Author Response · Authors · 2023-11-17
> **Response to Reviewer QeLY (2/2)**
>
> __Q: It remains unclear whether deep learning benchmarks such as N-BEATS, Autoformer, and FEDformer are only trained on designated training sets. Exploring whether similar performance improvements could be achieved by training these models on the pre-training set, using the same loss function, might offer deeper insights. This also helps determine the source of the proposed model's performance gains—whether from the architecture itself or mostly from expanded datasets.__
>
> A: We apologize for the potential source of confusion. Deep learning baselines follow the "from scratch" in-collection setting (Figure 2), where they are trained on the train region of the train-test set. We have updated the manuscript to more clearly reflect this information in Section 4.3.2.
>
> Thank you for the suggestion to include further results for other pre-trained models. Firstly, we would like to highlight that the Meta N-BEATS baseline is exactly as requested for N-BEATS above, and TS2Vec and CoST are also pre-trained on the pre-train set. We would also like to highlight that pre-training such models is resource intensive and we are unable to extensively scale and pre-train all models due to resource constraints. However, this is indeed a valuable addition and we have trained DeepAR and Autoformer models with equivalent hyperparams/size to the "Base" Masked Encoder model, with details as shown:
>
> |                 | Layers | d_model | d_ff | \# heads | d_kv | \# params |
> | --------------- | ------ | ------- | ---- | -------- | ---- | --------- |
> | DeepAR-Base     | 6      | 384     | \-   | \-       | \-   | 6.6m      |
> | Autoformer-Base | 6      | 384     | 1536 | 6        | 64   | 24.8m     |
> | Ours-Base       | 6      | 384     | 1536 | 6        | 64   | 10.7m     |
>
> As expected, we observe that scaling up DeepAR and Autoformer with pre-training does not yield an equivalent performance as the Masked Encoder architecture, indicating that a general, expressive Transformer architecture is more suitable and benefits more from pre-training and the zero-shot setting. We also note that these models sometimes underperform their vanilla non pre-trained counterparts -- this could be due to various factors. For example, it is possible that they are unable to generalize to overcome the distribution shift between the pre-train and train-test sets (as demonstrated in the above question), and also due to challenges involved in scaling up.
>
> ||azure2017||borg2011||ali2018||
> | --------------- | --------- | --------- | --------- | --------- | --------- | --------- |
> || sMAPE| CRPS| sMAPE|CRPS|sMAPE|CRPS|
> | Meta N-BEATS    | 0.120     | 0.116     | \-        | \- | \- | \- |
> | DeepAR-Base     | 0.216     | 0.163     | 0.066     | 0.149 | 0.240 | 0.053 |
> | Autoformer-Base | 0.171     | 0.165     | 0.187     | 0.235 | 0.266 | 0.065 |
> | Ours-Base       | **0.084** | **0.079** | **0.061** | **0.128** | **0.154** | **0.016** |

---

> ### Author Response · Authors · 2023-11-20
> **Update: Included results of additional baselines - pre-trained DeepAR and Autoformer**
>
> Dear Reviewer QeLY,
>
> Thank you for your patience, we have completed the experiments for pre-training DeepAR and Autoformer, and have updated our response to include these results. We hope that our rebuttal has managed to provide clarity and address any remaining concerns you may have had on our work, and that you have had a chance to review our rebuttal. We are eager to address any remaining concerns, or if our rebuttal has addressed all your concerns, we hope that you could kindly leave an update to reflect this. Thank you so much for your time and effort.

---

> ### Author Response · Authors · 2023-11-21
>
> Dear Reviewer QeLY,
>
> Thank you for taking the time and effort in providing a valuable review of our work thus far. As the discussion period is soon coming to a close, we hope that you have had the chance to review our rebuttal. We believe that our rebuttal has provided additional clarity and highlighted the strengths and contributions of our work. If our rebuttal has addressed your concerns, we hope that you could kindly leave an update to reflect this, and are more than willing to engage in further discussion if needed.

---

### Official Review · Reviewer_LRUs · 2023-11-01

**Soundness:** 3 good
**Presentation:** 3 good
**Contribution:** 1 poor
**Rating:** 3
**Confidence:** 4

**Summary:**

This scientific paper addresses the limited progress in applying pre-training and transfer learning to time series data. To bridge this gap, the authors introduce three large-scale time series forecasting datasets from the CloudOps domain, with the largest dataset containing billions of observations. This substantial dataset size allows for a comprehensive investigation into the effectiveness of pre-training and scaling for time series models. The paper establishes the groundwork for studying pre-training and scaling of time series models and identifies a promising architecture for the task. This architecture serves as a strong zero-shot baseline and exhibits further improvements with increased model and dataset size. The authors provide a benchmark comparing several classical and deep learning methods with their proposed pre-trained approach, demonstrating significant reduction in error on the largest dataset.

**Strengths:**

1. The manuscript is well-written and easy to follow. The authors thoroughly explain various setups, such as pretraining and fine-tuning, and make an attempt to define a taxonomy of time series data based on domains, collections, and individual time series.
2. The work introduces three large-scale time series forecasting datasets, which enable a deeper exploration of pretraining and transfer learning for time series models. The authors provide concrete details into the process of transforming raw data into useful time series data.
3. The paper goes on to examine various Transformer architectures for forecasting, conducts a comprehensive benchmarking analysis against classical and deep learning forecasting methods, and also compares their proposed pretraining approaches with existing methods. Additionally, the study investigates the impact of scaling in terms of model parameters and training data size on time series forecasting.

**Weaknesses:**

1. Lack of Novelty: The paper's approach is primarily centered on training a large transformer model on an extensive time series dataset, which may not offer a novel contribution to the field.
2. Limited Analysis of Transfer Learning: The study raises concerns regarding the assessment of transfer learning, as the model is trained on one dataset and tested on a similar one without sufficient information about their differences. This makes it challenging to interpret the extent of transfer learning within an in-collection setting. Additionally, the diversity of time series within the CloudOps datasets remains unclear, which could lead to potential overfitting issues, particularly for large models.
3. Inadequate Baseline Model Details: The paper lacks essential information about the baseline models, including their model size and training duration. For instance, while DeepAR emerges as the second best method in Table 5, it is uncertain if its number of parameters is comparable to the proposed transformer models. A comparison with DeepAR models of similar scale and training iterations would provide valuable insights.
4. Limited Improvements in Larger Models: The paper reveals that the gains achieved with "Large" and "xLarge" models are not significant when compared to the base model size. Particularly, the "xLarge" model, despite being over eight times larger than the base model, exhibits only marginal improvements. This suggests that the base model may already be overfitting the datasets.

**Questions:**

See weakness section.

---

> ### Author Response · Authors · 2023-11-17
> **Response to Reviewer LRUs (1/2)**
>
> Thank you for taking the time and effort to review our work. We hope that our responses below fully address your concerns and queries, and look forward to further discussion.
>
> __Q: Lack of novelty.__
>
> A: We address this in our response to all reviewers since this is a recurring issue amongst reviewers.
>
> __Q: Limited analysis of transfer learning.__
>
> A: Thank you for this valuable suggestion -- we have updated in our manuscript Appendix A.3, where we perform further data analysis to demonstrate the distribution shift between pre-train and train-test sets.
>
> As outlined in Appendix A.2, we ensure that the data split is done based on a *top level attribute*, ensuring that there is no overlap between the pre-train and train-test sets. Thus, we expect the time series patterns/distribution across pre-train and train-test sets to be different, making the in-collection pre-training setting to be meaningful and challenging. We verify this hypothesis in the new Appendix A.3, where we perform a (1) qualitative analysis, and a (2) quantitative analysis.
>
> 1. Qualitatively, we show in Figure 7, based on histogram plots comparing the (a) train region vs pre-train set, and (b) train region vs train-test region, that the gap in (a) is larger than the gap in (b).
> 2. Quantitatively, we obtain a measure of distribution shift between the pre-train set and train-test set. We do so by first encoding the time series into a feature vector, comprising of 7 representative features as suggested by [1] -- spectral entropy, strength of trend (hourly and daily), strength of seasonality (hourly and daily), first-order autocorrelation, and the optimal box-cox transformation parameter. Then, we use the Wasserstein distance to measure the distribution difference between pre-train and train-test distributions. Table 7 demonstrates, by comparing with a a baseline of the Wasserstein distance between two random subsets of the pre-train set, that there is indeed a difference in distribution between the pre-train and train-test sets.
>
> |                         | azure2017     | borg2011      | ali2018       |
> | ----------------------- | ------------- | ------------- | ------------- |
> | Pre-train vs Train-test | 5.26          | 1.69          | 9.81          |
> | Pre-train               | 0.160 (0.013) | 0.077 (0.024) | 0.361 (0.024) |
> | Train-test              | 0.802 (0.27)  | 0.296 (0.098) | 0.545 (0.268) |
>
> Regarding dataset diversity, there are several aspects to discuss. On the level of data attributes, as seen in Table 6 in Appendix A.1, azure2017 measures 1 target variable (CPU utilization) whereas borg2011 and ali2018 measures 2 target variables (CPU utilization and memory usage). Furthermore, each dataset measures unique auxiliary covariates, for example, azure2017 has unique static real features such as virtual core count, memory size, and deployment size.
>
> Next, we can do a qualitative comparison based on time series visualizations in Appendix F. azure2017 has high frequency cyclical patterns, whereas borg2011 has slightly lower frequency patterns. On the other hand, ali2018 has very low frequency seasonality, and contains significantly more trend information.
>
> Finally, similar to the analysis on distribution shift, we can perform a quantitative evaluation of the diversity of the datasets. Using the same approach as before, we measure the distance between the distribution of datasets using the Wasserstein distance (we have included this as Appendix A.3.2, including a scatter plot of the features in low-dimensional feature space generated by PCA).
>
> |               | azure2017    | borg2011      | ali2018       |
> | ------------- | ------------ | ------------- | ------------- |
> | azure2017 | 0.802 (0.27) | 42.86         | 20.85         |
> | borg2011      | 42.86        | 0.296 (0.098) | 10.77         |
> | ali2018       | 20.85        | 10.77         | 0.545 (0.268) |
>
> [1] Godahewa, Rakshitha, et al. "Monash time series forecasting archive." arXiv preprint arXiv:2105.06643 (2021).

---

> ### Author Response · Authors · 2023-11-17
> **Response to Reviewer LRUs (2/2)**
>
> __Q: Inadequate baseline model details.__
>
> A: Due to space limitations, details of baseline models are included in Appendix E. The training procedure follows the "fine-tuning/training from scratch" procedure as outlined in Section 4.1. Critical hyparameter search range which determine model sizes can be found in Table 10 (Appendix E).
>
> One potential source of confusion we would like to highlight, is that the baselines (apart from the pre-trained baselines, i.e. TS2Vec, CoST, and Meta N-BEATS) follow the "from scratch" in-collection setting (see Figure 2). We have updated the manuscript to more clearly reflect this information in Section 4.3.2. The reason for this experiment choice is that (1) we want to evaluate a strong in-collection zero-shot forecaster with the classical "train from scratch" paradigm, and (2) pre-training these large models is resource intensive and we are unable to extensively scale and pre-train all models.
>
> That being said, the reviewer has raised a critical point regarding comparison of other models, e.g. DeepAR, of similar scale and training approach. We currently have one baseline - Meta N-BEATS which does so, and have included new baselines, scaled + pre-trained DeepAR and Autoformer models with equivalent hyperparams/size to the "Base" Masked Encoder model, with details as shown:
>
> |                 | Layers | d_model | d_ff | \# heads | d_kv | \# params |
> | --------------- | ------ | ------- | ---- | -------- | ---- | --------- |
> | DeepAR-Base     | 6      | 384     | \-   | \-       | \-   | 6.6m      |
> | Autoformer-Base | 6      | 384     | 1536 | 6        | 64   | 24.8m     |
> | Ours-Base       | 6      | 384     | 1536 | 6        | 64   | 10.7m     |
>
> As expected, we observe that scaling up DeepAR and Autoformer with pre-training does not yield an equivalent performance as the Masked Encoder architecture, indicating that a general, expressive Transformer architecture is more suitable and benefits more from pre-training and the zero-shot setting. We also note that these models sometimes underperform their vanilla non pre-trained counterparts -- this could be due to various factors. For example, it is possible that they are unable to generalize to overcome the distribution shift between the pre-train and train-test sets (as demonstrated in the above question), and also due to challenges involved in scaling up.
>
> ||azure2017||borg2011||ali2018||
> | --------------- | --------- | --------- | --------- | --------- | --------- | --------- |
> || sMAPE| CRPS| sMAPE|CRPS|sMAPE|CRPS|
> | Meta N-BEATS    | 0.120     | 0.116     | \-        | \- | \- | \- |
> | DeepAR-Base     | 0.216     | 0.163     | 0.066     | 0.149 | 0.240 | 0.053 |
> | Autoformer-Base | 0.171     | 0.165     | 0.187     | 0.235 | 0.266 | 0.065 |
> | Ours-Base       | **0.084** | **0.079** | **0.061** | **0.128** | **0.154** | **0.016** |
>
> __Q: Limited improvements in larger models.__
>
> A: The reviewer has indeed pointed out an accurate observation. Instead of a weakness, this is one of the findings presented in our empirical analysis of scaling up time series models. Motivated by these results, we performed a more in-depth analysis. In Section 4.4, Figure 5, we observe that as we increase training iterations/number of training samples, the azure2017 dataset reveals a clear trend which starts to plateau. However, there is no such relationship for the borg2011 and ali2018 dataset, which critically, are 0.5x and 0.1x the size of the azure2017 dataset. We indeed come to the same conclusion as the reviewer, highlighting this fact as a key finding in Section 4.4, and providing further evidence in Appendix G by plotting even more fine-grained plots of validation error across different training iterations.

---

> ### Author Response · Authors · 2023-11-20
> **Update: Included results of additional baselines - pre-trained DeepAR and Autoformer**
>
> Dear Reviewer LRUs,
>
> Thank you for your patience, we have completed the experiments for pre-training DeepAR and Autoformer, and have updated our response to include these results. We hope that our rebuttal has managed to provide clarity and address any remaining concerns you may have had on our work, and that you have had a chance to review our rebuttal. We are eager to address any remaining concerns, or if our rebuttal has addressed all your concerns, we hope that you could kindly leave an update to reflect this. Thank you so much for your time and effort.

---

> ### Author Response · Authors · 2023-11-21
>
> Dear Reviewer LRUs,
>
> Thank you for taking the time and effort in providing a valuable review of our work thus far. As the discussion period is soon coming to a close, we hope that you have had the chance to review our rebuttal. We believe that our rebuttal has provided additional clarity and highlighted the strengths and contributions of our work. If our rebuttal has addressed your concerns, we hope that you could kindly leave an update to reflect this, and are more than willing to engage in further discussion if needed.

---

### Author Response · Authors · 2023-11-17
**General response to all reviewers (1/2)**

Thank you to all reviewers for taking the time and effort to read through our work, and for the insightful comments and feedback. We have made changes and added further analyses based on the valuable feedback, which is highlighted in red in the updated manuscript. Below is a summary of the changes made:

1. Added further details of deep learning baselines in Section 4.3.2.
2. New section (Appendix A.3) performing data analysis to provide further insight to the (1) distribution shift between the pre-train and train-test sets, and (2) diversity of the 3 datasets.
3. Additional baseline results of DeepAR and Autoformer on the same scale as the "Base" size, trained on the pre-training datasets.

We also summarize some of the positive sentiments that the reviewers had about our work:

With regards to __presentation__, reviewers found that
* *the manuscript is well-written and easy to follow* (LRUs)
* *it makes an effort on having well structured terminology in these new and developing field* (boWu)

Regarding the __soundness__ of our work, reviewers
* *thoroughly explain various setups, such as pretraining and fine-tuning* (LRUs)
* *provide concrete details into the process of transforming raw data into useful time series data* (LRUs)

Finally, regarding __contribution__,
* *introduces the first pre-trained time-series model specifically for cloud operation domains* (QeLY)
* *conducts a comprehensive benchmarking analysis against classical and deep learning forecasting methods* (LRUs)
* *provide an interesting analysis on variants of transformer models, providing further insights on what approaches are more promising in the future* (boWu)
* *This contribution is relevant in the field of time series forecasting, as there is certain lack of this kind of datasets, holding back the progress in the topic of pretrained models or LLM-based forecasting models* (boWu)

---

> ### Author Response · Authors · 2023-11-17
> **General response to all reviewers (2/2)**
>
> Despite highlighting the above contributions, there was still a sentiment that our work did not have __sufficient contribution in terms of novelty__, and a recurring theme in the reviews was that we only performed empirical analyses, introduced new datasets, and a benchmark. We fully agree that our work is not aiming to introduce any novel architectures or model design. Yet, we respectfully argue that this is not the only approach to make a significant contribution to the field -- empirical analyses and datasets and benchmarks are becoming increasingly important in data-driven fields, such as ours.
>
> We hope to encourage the reviewers to view our work in the lens of an empirical study cum dataset and benchmark paper -- not as a paper which attempts to present large transformers trained on extensive time series datasets as a novel method. For ease of reading, we summarize and reiterate our key contributions:
>
> 1. New large-scale dataset + comprehensive benchmarking.
>
>     * __Why is this important?__
>     * Foundation models are the new holy grail across all data-driven fields, no less in time series forecasting. In ICLR 2024 alone, we see many submissions attempting to build foundation models for time series forecasting [1,2,3,4,5]. However, we note that these works are heavily limited by the availability of open time series datasets, training their models on at most __millions of observations__. Compared to fields like NLP, which trains on __trillions of tokens__, there is still much work to be done for time series.
>     * Size is not the only critical factor in terms of building datasets for foundation models. Most open time series forecasting datasets are concerned with the domains of energy, traffic, weather, and sales. We introduce __diversity__, with the largest CloudOps time series dataset.
>
> 2. Verify that time series models indeed benefit from scaling (highly contended issue in time series forecasting [6,7,8]).
>
>     * __Why is this important?__
>     * As a result of dataset limitations, conflicting evidence has been presented in the time series forecasting literature [6,7,8] regarding the value of scale and expressive models. Evidence has been shown that lightweight models outperform expressive Transformer-based architectures and data scale is not a limiting factor. These experiments were performed on arguably small dataset sizes -- the external validity (extrapolating these findings to larger scales) is tenuous.
>     * We present definitive evidence that time series can indeed benefit from further scaling in terms of dataset and model sizes.
>
> 3. Present a strong candidate (Transformer-based) architecture for further studies, selected via a series of ablations.
>
>     * __Why is this important?__
>     * Transformers have emerged as the defacto architecture to pre-train and develop foundation models. However, the specific style and details (e.g. Encoder-only, Decoder-only, etc.) varies between fields (even within NLP, there is still contention between Decoder-only vs Encoder-Decoder). We establish that the Masked Encoder variant is a strong candidate for time series forecasting.
>
> In this viewpoint, while our work may not be introducing novel designs, we argue that it still constitutes an important and significant contribution to the field, building up the groundwork to bring time series forecasting towards the era of foundation models.
>
> [1] [Time Series Modeling at Scale: A Universal Representation Across Tasks and Domains](https://openreview.net/forum?id=SZErAetdMu)
>
> [2] [DAM: A Foundation Model for Forecasting](https://openreview.net/forum?id=4NhMhElWqP)
>
> [3] [PEMs: Pre-trained Epidemic Time-Series Models](https://openreview.net/forum?id=DL7JWbdGr3)
>
> [4] [Large Pre-trained time series models for cross-domain Time series analysis tasks](https://openreview.net/forum?id=KJ1w6MzVZw)
>
> [5] [TimelyGPT: Recurrent Convolutional Transformer for Long Time-series Representation](https://openreview.net/forum?id=2sCcTMWPc2)
>
> [6] Spyros Makridakis, Evangelos Spiliotis, and Vassilios Assimakopoulos. Statistical and machine learning forecasting methods: Concerns and ways forward. PloS one, 13(3):e0194889, 2018.
>
> [7] Ailing Zeng, Muxi Chen, Lei Zhang, and Qiang Xu. Are transformers effective for time series forecasting? In Proceedings of the AAAI conference on artificial intelligence, volume 37, pp. 11121–11128, 2023.
>
> [8] Zhijian Xu, Ailing Zeng, and Qiang Xu. Fits: Modeling time series with 10k parameters. arXiv preprint arXiv:2307.03756, 2023.

---

### Meta-Review · Area_Chair_7xKv · 2023-12-07

**Metareview:**

The paper presents an empirical analysis for pretraining a transformer model on a set of three large cloudops timeseries datasets. The authors benchmark  their pre-trained models and show that their models obtains zero-shot performance that is better than trained-from-scratch deep forecasting baselines.
The paper is addressing a timely and relevant topic in time-series forecasting around pre-trained foundation models and transfer-learning in time-series. Also, the addition of new large-scale time series datasets will be very useful for future research on timeseries foundation models. However the reviewers expressed a major concern (that AC agrees with) that all the evaluations datasets are from the CloudOps domain, which brings into question whether the the zero-shot/transfer-learning findings in the paper will  be applicable across diverse datasets and domains.
I would urge the authors to strengthen the paper with evaluations on more diverse datasets from multiple domain.

**Justification For Why Not Higher Score:**

The evaluation results and zero-shot/transfer-learning claims in this paper is significantly weakened by the fact that the authors restricted both pretraining and evaluation datasets to those from a single CloudOps domain, without considering at more diverse evaluation datasets.

**Justification For Why Not Lower Score:**

N/A

---

### Decision · Program_Chairs · 2024-01-16

Reject